# Direct time-resolved observation of surface-bound carbon dioxide radical anions on metallic nanocatalysts

Zhiwen Jiang [1,2], Carine Clavaguéra [2], Changjiang Hu[3], Sergey A. Denisov[2], Shuning Shen[3], Feng Hu[3], Jun Ma [1] ✉ & Mehran Mostafavi [2] ✉

Time-resolved identification of surface-bound intermediates on metallic nanocatalysts is imperative to develop an accurate understanding of the elementary steps of $CO_2$ reduction. Direct observation on initial electron transfer to $CO_2$ to form surface-bound $CO_2^{\cdot-}$ radicals is lacking due to the technical challenges. Here, we use picosecond pulse radiolysis to generate $CO_2^{\cdot-}$ via aqueous electron attachment and observe the stabilization processes toward well-defined nanoscale metallic sites. The time-resolved method combined with molecular simulations identifies surface-bound intermediates with characteristic transient absorption bands and distinct kinetics from nanosecond to the second timescale for three typical metallic nanocatalysts: Cu, Au, and Ni. The interfacial interactions are further investigated by varying the important factors, such as catalyst size and the presence of cation in the electrolyte. This work highlights fundamental ultrafast spectroscopy to clarify the critical initial step in the $CO_2$ catalytic reduction mechanism.

The conversion of planet-warming carbon dioxide ($CO_2$) into valuable chemicals or fuels is considered one of the appealing methods for storing renewable energy and facilitating sustainable carbon cycles[1,2]. However, the high activation energy associated with the change in orbital hybridization and geometry of $CO_2$ upon the first electron attachment, the generation of $CO_2$ radical anion ($CO_2^{\cdot-}$), makes its reduction a significant challenge. As a result, the successful operation depends heavily on the viable catalysts that can accelerate the reaction. In most electrolytic, photolytic, and radiolytic processes, metal-based catalysts primarily contribute to $CO_2$ transformations[3–6]. Active metals can be categorized into several groups based on their product distribution. For example, Ag and Au predominately yield CO, while Ni and Fe exclusively produce $H_2$ with low activity, and Cu has an unusual capacity to synthesize hydrocarbon/multi-carbon products[7–9]. These findings raise questions about the importance of reaction intermediates binding at the interface on the efficiency. Although great progress has been made in understanding the underlying surface chemistry, the reaction sequence remains incomplete, partly due to

the lack of direct experimental evidence for unraveling the fast dynamics of $CO_2^{\cdot-}$ during the initial stages of complex multi-electron processes. On the other hand, advances in synthetic methodology have made well-controlled metallic nanoparticles and nanoclusters available, and these nanostructured materials exhibit better performance in $CO_2$ conversion and lower cost. Strategies such as surface engineering, morphology control, and composition manipulation have been developed to enhance the efficiency of $CO_2$ activation[10–12]. In this regard, the characterization of $CO_2^{\cdot-}$ coordination complex with metal centers could provide a basic descriptor to establish the structure-property correlation, where any transient state changes with variables of interest at nanoscale spatial resolution could be detected. However, detecting these active sites directly from experiments has been impossible thus far[13–16].

The existence of $CO_2^{\cdot-}$ on metal electrodes, such as Au, Cu, and Ag, has been proposed based on the experimental Tafel slope analysis[17–21]. Besides, the critical role of $CO_2^{\cdot-}$ formed by hydrated electrons ($e_{aq}^-$) attachment has recently been recognized by plasma or

[1]School of Nuclear Science and Technology, University of Science and Technology of China, 230026 Hefei, Anhui, P. R. China. [2]Université Paris-Saclay, CNRS, Institut de Chimie Physique, 91405 Orsay, France. [3]Department of Materials Science and Technology, Nanjing University of Aeronautics and Astronautics, 211106 Nanjing, P. R. China. ✉e-mail: majun0502@ustc.edu.cn; mehran.mostafavi@universite-paris-saclay.fr

radiation-driven catalytic and nanodiamond-assisted photolytic $CO_2$ reduction[22–24]. Different catalytic systems have several rate-determining steps (RDS), such as initial electron transfer to $CO_2$ to form surface-bound $CO_2^{\bullet-}$ radicals, the first proton-coupled electron transfer, or the reduction of other bound intermediates[17–21]. Due to the large structural reorganization of the bound radical anion, the radical anion $CO_2^{\bullet-}$ formed by the first electron reduction occurs at very negative potentials. Even though $CO_2^{\bullet-}$ formation lacked experimental evidence, many studies suggested that the process is likely to be one of the critical RDS for $CO_2$ reduction[7,9,15]. If the $CO_2^{\bullet-}$ radicals stabilization on catalytic surfaces is kinetically non-favorable, the overall activity for $CO_2$ transformation will be low. This hypothesis is further supported by theoretical studies that developed a contemporary model for various *$CO_2^{\bullet-}$ coordination structures, including *COOH or *OCOH, uncovering the selective electrocatalytic pathway toward either CO or formate[25–30]. In situ surface-enhanced Raman scattering has reported the presence of this surface species on an operating Cu surface, but the dynamic resolution remains unresolved[14,26,31,32]. For nanocatalysts, operando X-ray absorption and photoelectron spectroscopy are sensitive to surface oxidation evolution and to the accurate active site of metals such as Cu, Pd, Zn, and Sn during $CO_2$ reduction[25,27,33,34]. Few studies have been dedicated to the initial step of $CO_2$ reduction at a nanoscale catalytic surface, owing to the challenges of forming well-defined nanocatalysts and $CO_2^{\bullet-}$ intermediate. Additionally, unlike carbon monoxide, which is frequently used as a mechanistic probe, $CO_2^{\bullet-}$ radicals decay rapidly in aqueous solutions, with a short lifetime of microseconds. The existing operando techniques are limited to the resolution of seconds or sub-seconds range, which cannot access the first elementary reactions occurring at nanosecond timescale.

For initial $CO_2$ activation, pulse radiolysis provides a time-resolved prerequisite to insight into the general transient absorption kinetics of $CO_2^{\bullet-}$. Radiolysis is often referred to as "an electrolysis process without electrodes". The high-energy radiation (X-rays/accelerated e⁻) with sufficient energy ejects electrons directly from the water. Unlike electrochemical or photochemical reduction, where electrons are provided by the external electric field or photo-induced carriers (e⁻, h⁺) separation in semiconductor materials, the radiation-driven approach produces electrons in bulk solutions via water ionization, known as the $e_{aq}^-$ with characteristic transient absorption at 715 nm. With the standard potential of −2.87 $V_{SHE}$, $e_{aq}^-$ can easily overcome the first $CO_2$ reduction step barrier[35]. The picosecond pulse radiolysis coupled with transient absorption spectroscopy (ELYSE platform, Paris-Saclay University) could form $e_{aq}^-$ in $CO_2$-saturated aqueous conditions within 7 ps, and $e_{aq}^-$ reacts directly and quantitatively with $CO_2$ forming $CO_2^{\bullet-}$ radicals with an almost diffusion-controlled rate ($8 \times 10^9 M^{-1} s^{-1}$) within 10 ns. Until now, the current in situ/operando spectroscopic measurements on $CO_2$ reduction were performed with limited time resolution in a few studies with sub-second time resolution based on infrared, Raman, MS, and XAS[30,36–38]. Compared to these spectroscopic techniques based on molecular vibration or photoelectron excitation, transient absorption profiles can disclose the electronic transition of surface-bound intermediates on metallic nanocatalysts with the nanosecond time resolution, which allows for the direct observation of the reaction between $CO_2^{\bullet-}$ and nanoparticles (NPs) as well as their stability kinetics on the surfaces. The method avoids external potential supported catalysts perturbation and the complexity associated with processes involving photosensitizers or electrodes. Therefore, pulse radiolysis measurements allow us to gain critical insights into the first elementary step of $CO_2$ reduction, such as the intermediate binding and transformation, as well as charge delocalization on nanoscale metallic surfaces. This methodology would set up a model to investigate the factors in operating conditions, such as cations in electrolytes and the nano-size effect of catalysts, which are important but under intense debate in $CO_2$ reduction because the existing in situ spectroscopic measurements often deal with combined signals caused by various possible consequences including solution effects, competing reactions, and surface structure evolution or catalysts oxidation.

Herein, by using pulse radiolysis[39,40], we present direct transient spectroscopic evidence of $CO_2^{\bullet-}$ radical stabilization process on the surface of nanoscale catalysts, leading to the formation of surface-bound $CO_2^{\bullet-}$. The characteristic absorption spectrum of surface species and density functional theory (DFT) simulations revealed the coordinated structure and interactions between metal clusters and $CO_2^{\bullet-}$ radicals. The Tafel analysis and electrochemical catalytic performance with the same NPs used in the pulse radiolysis observations are also performed. As a result, our distinct transient kinetics on various well-defined metal (Au, Cu, Ni) nanoparticles (NP), which are commonly included in electrocatalysis, provided fundamental insights to revisit the activity and selectivity of metal catalysts in $CO_2$ reduction.

## Results

### Reaction kinetics of $CO_2^{\bullet-}$ radicals in bulk solution

Supplementary Figure 1a presents the transient absorption profiles of $CO_2$ radical formation in $CO_2$-saturated formate aqueous solution in the absence of any catalyst. Immediately following the electron pulse, the typical broad band in the 400–720 nm range corresponds to $e_{aq}^-$. The appearance of a new absorption band below 420 nm on the nanosecond timescale, which is associated with $e_{aq}^-$ decay, indicated the rapid formation of $CO_2^{\bullet-}$ radicals via $e_{aq}^-$ reduction ($e_{aq}^- + CO_2 \rightarrow CO_2^{\bullet-}$, $k_1 = 8.2 \times 10^9 M^{-1} s^{-1}$). Meanwhile, 0.1 M formate was used to convert $^{\bullet}OH$ and $^{\bullet}H$ into $CO_2^{\bullet-}$ radicals ($^{\bullet}OH + HCOO^- \rightarrow CO_2^{\bullet-} + H_2O$, $k_2 = 4.1 \times 10^9 M^{-1} s^{-1}$; $^{\bullet}H + HCOO^- \rightarrow CO_2^{\bullet-} + H_2$, $k_3 = 2.1 \times 10^9 M^{-1} s^{-1}$), thereby converting all water radiolysis radicals into $CO_2^{\bullet-}$ radicals. Subsequently, $CO_2^{\bullet-}$ radicals underwent a recombination reaction ($CO_2^{\bullet-} + CO_2^{\bullet-} \rightarrow C_2O_4^{2-}$, $k_4 = 1.9 \times 10^9 M^{-1} s^{-1}$) and lived for approximately 10 μs under the experimental conditions (Supplementary Fig. 1b–d). These data depicted a landscape of the formation and decay process of $CO_2^{\bullet-}$ radicals in our conditions and in the absence of metal catalysts, in line with those reported in the literature[41,42].

### The effect of nanoscale metal catalysts (Au/Cu/Ni) on the $CO_2^{\bullet-}$ radical reactivity

To elucidate the $CO_2^{\bullet-}$ radical reactivity with nanocatalysts, we synthesized three typical (Au/Cu/Ni) metal NP by an established radiolytic reduction method with surfactants to regulate the size[43,44] (details in Methods and Supplementary Fig. 2). In addition to our time-resolved measurements, we performed electrochemical analysis on these three NPs. The obtained Tafel slope for CO production with (Au/Cu/Ni) NPs as-synthesized in our work agreed with previous reports[17–21]. The distinct activity indicates a rate-determining initial e⁻ transfer from $CO_2$ to $CO_2^{\bullet-}$ intermediates (Supplementary Fig. 3), excluding the possible RDS relative to protonation and the coupling reaction of $CO_2^{\bullet-}$ intermediates. The transient absorption profiles of $CO_2^{\bullet-}$ radicals in the presence of these nanoscale metal catalysts from nanoseconds to microseconds are shown in Fig. 1. Figure 1a illustrates the transient kinetics of $CO_2^{\bullet-}$ at 350 nm in various concentrations of Au NP solution within 700 ns. Obviously, Au NP suppresses the decay of $CO_2^{\bullet-}$ and even induces the growth of a new absorption band extending to nearly 520 nm during 7 μs as the concentration of Au rises to 0.5 mM. The optical spectral observations suggest that the gradual introduction of Au NPs inhibited the recombination of $CO_2^{\bullet-}$ radicals, promoting the formation of new surface species instead of oxalate (Fig. 1b).

To investigate the origin of the absorbing species, we performed control experiments and analyzed the kinetics of the supernatant of Au suspensions (solution prepared for pulse radiolysis measurements but underwent centrifugation to remove Au NP). Our results ruled out the possibility that the new absorption band is caused by the reaction of $CO_2^{\bullet-}$ with any other components in the solution (Supplementary

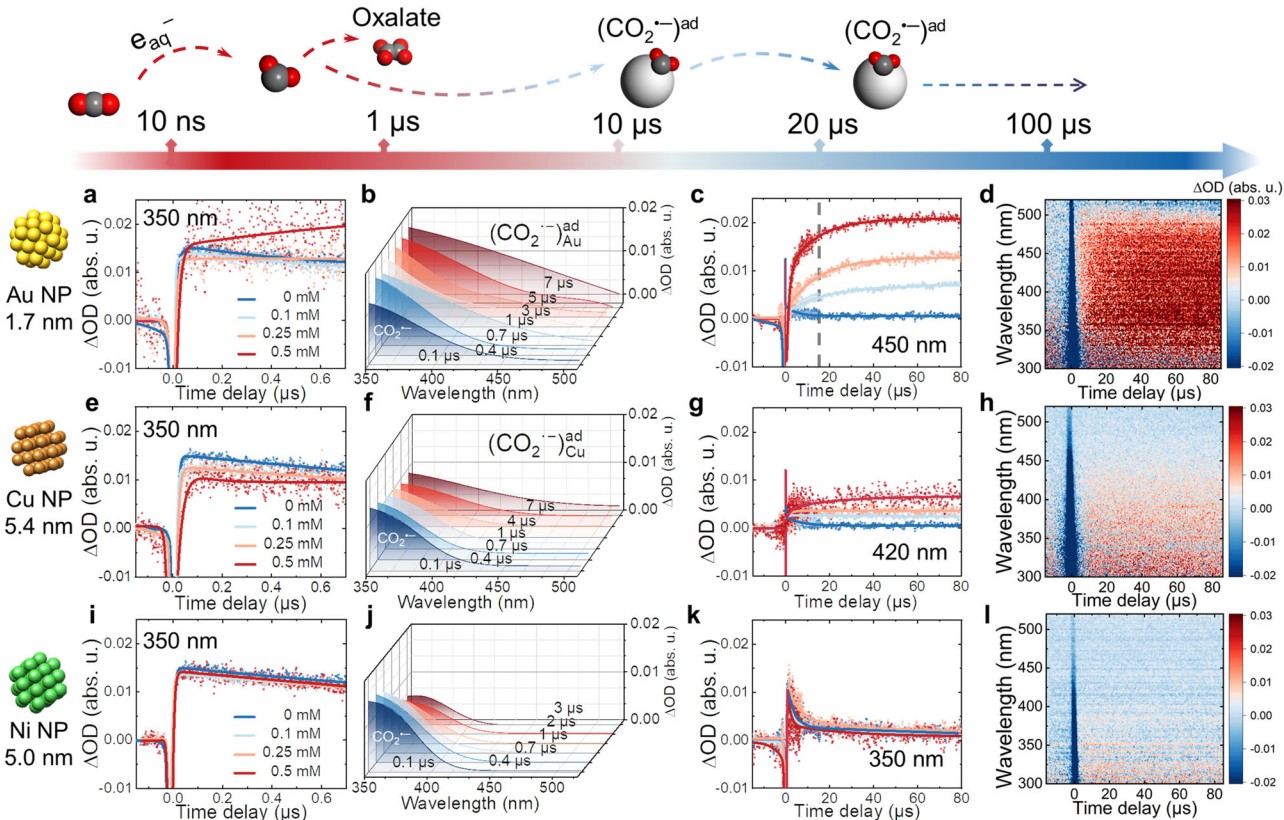

**Fig. 1 | Time-resolved absorption of $CO_2^{\cdot-}$ radical stabilization process with different metal NP. a–d** Transient kinetics at 350 nm within 700 ns as a function of Au concentration (**a**), fitted transient absorption spectra within 7 μs in 0.5 mM Au solution (**b**), transient kinetics at 420 nm as a function of Au concentration (**c**), and transient absorption matrix (**d**) within 80 μs in 0.5 mM Au solution. **e–h** Transient kinetics at 350 nm within 700 ns as a function of Cu concentration (**e**), fitted transient absorption spectra within 7 μs in 0.5 mM Cu solution (**f**), transient kinetics

at 450 nm as a function of Cu concentration (**g**), and transient absorption matrix (**h**) within 80 μs in 0.5 mM Cu solution. **i–l** Transient kinetics at 350 nm within 700 ns as a function of Ni concentration (**i**), fitted transient absorption spectra within 7 μs in 0.5 mM Ni solution (**j**), transient kinetics at 350 nm as a function of Ni concentration (**k**), and transient absorption matrix (**l**) within 80 μs in 0.5 mM Ni solution. Source data are provided as a Source Data file.

Fig. 4a–e) than Au NP. To ensure the signals are from the surface-bound $CO_2^{\cdot-}$ on Au NPs, we performed an additional control experiment of the Au NPs system by replacing $CO_2$ with Ar. In these conditions, the only reducing species is $e_{aq}^-$ since ·OH radical is scavenged by tert-butanol. The results clearly show the absence of the absorption band as in the case of $CO_2$-saturated solutions (Supplementary Fig. 4f, g). Additionally, we found that the absorption intensity of the new species correlates positively with the concentration of Au NP (Supplementary Fig. 5d). These experimental data lead us to conclude that the new species absorbing in the visible and UV range corresponds to $CO_2^{\cdot-}$ radicals adsorbed on Au surfaces ($CO_2^{\cdot-} + Au \rightarrow (CO_2^{\cdot-})_{Au}^{ad}$), and the broader absorption band compared to that of free $CO_2^{\cdot-}$ is attributed to the more electron delocalization of $(CO_2^{\cdot-})_{Au}^{ad}$ on Au surfaces. Considering the low concentration of Au NP, such kinetics suggest the effective occurrence of $CO_2^{\cdot-}$ radical stabilization on Au surfaces, outcompeting the fast recombination reaction.

The transient kinetics at 450 nm in the range of 80 μs offers insights into the stabilization process since it corresponds to almost exclusive absorption for $(CO_2^{\cdot-})_{Au}^{ad}$ with respect to free $CO_2^{\cdot-}$ radicals in solutions (Fig. 1c). The formation kinetics of $(CO_2^{\cdot-})_{Au}^{ad}$ is accelerated significantly with Au NPs concentration during the first 20 μs. After 20 μs, free $CO_2^{\cdot-}$ radicals are nearly eliminated by the recombination reaction, and the stabilization on Au surfaces is complete. However, the absorption at 450 nm continues to grow until 80 μs, combined with the slight broadening of the spectra, suggesting an optimization process of the adsorbed $CO_2^{\cdot-}$ radicals on Au surfaces (Supplementary Fig. 5a). After a few 100 microseconds, the spectral band shape hardly

changes with time. The protonation reaction can also accompany this stabilization process by forming $HCO_2^{\cdot}$ on the surface of NPs. Nevertheless, as the pKa of $CO_2^{\cdot-}$ is 2.8, the possibility of the $CO_2^{\cdot-}$ protonation is limited. These critical features demonstrate that $CO_2^{\cdot-}$ radicals stabilize on Au surfaces swiftly within 20 μs, after which the coordination state progressively orientates and sustains for more than 80 μs (Fig. 1d).

Next, we extended our studies to nanoscale Cu and Ni. In both cases, we pay attention to avoiding any oxidized sites on the surface NPs by performing all experiments in reducing conditions. As shown in Fig. 1e, compared with Au, the transient kinetics at 350 nm displays no growth during the first 700 ns, even at the highest concentration of Cu (0.5 mM). However, within 10 μs, a distinct species issued from the $CO_2^{\cdot-}$ radicals in solution is identified ($CO_2^{\cdot-} + Cu \rightarrow (CO_2^{\cdot-})_{Cu}^{ad}$) through a detailed comparison of the transient kinetics at 350 and 420 nm (Supplementary Fig. 6). The transient absorption in the Cu system also evidences the formation of $(CO_2^{\cdot-})_{Cu}^{ad}$ intermediates, which shows a characteristic absorption band stretching to around 420 nm (Fig. 1f). By comparing the kinetics for solutions containing the same size of Au (5.3 nm) and Cu (5.4 nm) NPs (Supplementary Fig. 7), within 80 μs, the kinetics and intensity of $CO_2^{\cdot-}$ radicals are similar due to the almost identical surface areas. However, after 80 μs, surface-bound $CO_2^{\cdot-}$ radicals exhibited accelerated decay on Cu NPs yet still increased on Au NPs (Supplementary Fig. 5b, e). These diverging kinetics between surface-bound $CO_2^{\cdot-}$ radicals on Au and Cu NPs at identical sizes suggest distinct stabilization processes, which may correlate with catalytic $CO_2$ activity and selectivity.

In the Ni case, $CO_2^{\bullet-}$ radicals show almost the identical transient kinetics as in nanocatalyst-free solution, with nearly negligible concentration dependency (Fig. 1i–l). At the 100 μs timescale, almost no absorption is detected with Ni (Supplementary Fig. 5c, f), underlying no surface $CO_2^{\bullet-}$ radicals on Ni NP are formed. According to previous literature[8,45], Ni is considered to exhibit a relatively low $CO_2RR$ catalytic activity due to hydrogen evolution reaction (HER) activity and strong binding capabilities with CO intermediates. Nevertheless, our results on $CO_2^{\bullet-}$ radical kinetics suggest that the activity of metal sites can be predetermined, well before the binding of intermediates from $CO_2$ dissociation.

To probe the structure evolution of surface-bound $(CO_2^{\bullet-})^{ad}$ radicals, we extended the observation of transient kinetics to the sub-millisecond range. Notably, $(CO_2^{\bullet-})^{ad}$ radicals on both Au and Cu surfaces maintain their specific spectral features even after 800 μs, with intensity dependent on the concentration (Fig. 2a and Supplementary Fig. 8). We observe that the transient kinetics of $(CO_2^{\bullet-})_{Au}^{ad}$ tend to plateau after 200 μs, whereas $(CO_2^{\bullet-})_{Cu}^{ad}$ shows a moderate decay (Fig. 2b–f). However, in the presence of Au NP with the same size as Cu, the decay of $(CO_2^{\bullet-})_{Au}^{ad}$ radicals occur as well (Supplementary Fig. 9), which reveals that the presence of larger NP results in a higher concentration of $CO_2^{\bullet-}$ radicals per particle, and the reactivity of surface-bound $(CO_2^{\bullet-})^{ad}$ may be altered to occur more readily. However, no new species are observed in the 300–720 nm range; only the moderate decay represents the possible next slow elementary transformation of $(CO_2^{\bullet-})^{ad}$ radicals.

In addition, the increased bleaching of the surface plasmon absorption band with time delay is found (Fig. 2d, g). Typically, such bleaching occurs in the femtoseconds or picoseconds range via electron thermalization and electron-phonon relaxation, which seems incompatible with our results[46–48]. We suggested that the rise in the bleaching signal with Au content (Fig. 2d), coupled with the concerted kinetics of $(CO_2^{\bullet-})^{ad}$, implies that $(CO_2^{\bullet-})_{Au}^{ad}$ radicals affect surface electrons relaxation and may promote a charge transfer from $(CO_2^{\bullet-})_{Au}^{ad}$ radicals to Au surfaces, resulting in the bleaching of the surface plasmon absorption band. This constitutes another piece of evidence to corroborate our observation of $CO_2^{\bullet-}$ radical stabilization at the nanocatalyst surfaces as the first elementary step in $CO_2RR$.

More importantly, we followed the next step of the elementary reaction of surface-bound $CO_2^{\bullet-}$ on the subsecond timescale. In the case of Cu NPs, we observe the formation of another intermediate following the decay of the first species at 0.05 s, which has an extended absorption band from 300 to 600 nm. This new surface species can be observed up to 900 ms, which is the limitation of observation (Fig. 3a). Moreover, the absorption spectra continue changing within 900 ms, which may indicate the multistep reaction of surface-bound species or the evolution of surface coordination structure. Due to the characteristic absorption band between 300 and 600 nm of the newly observed species, we exclude any stable steady-state product, such as oxalate, formate, or CO. It could be attributed to a new transient species with conjugated structure involving two $CO_2^{\bullet-}$ radicals as the precursor for CO and formate, already suggested in nanocatalyst-free

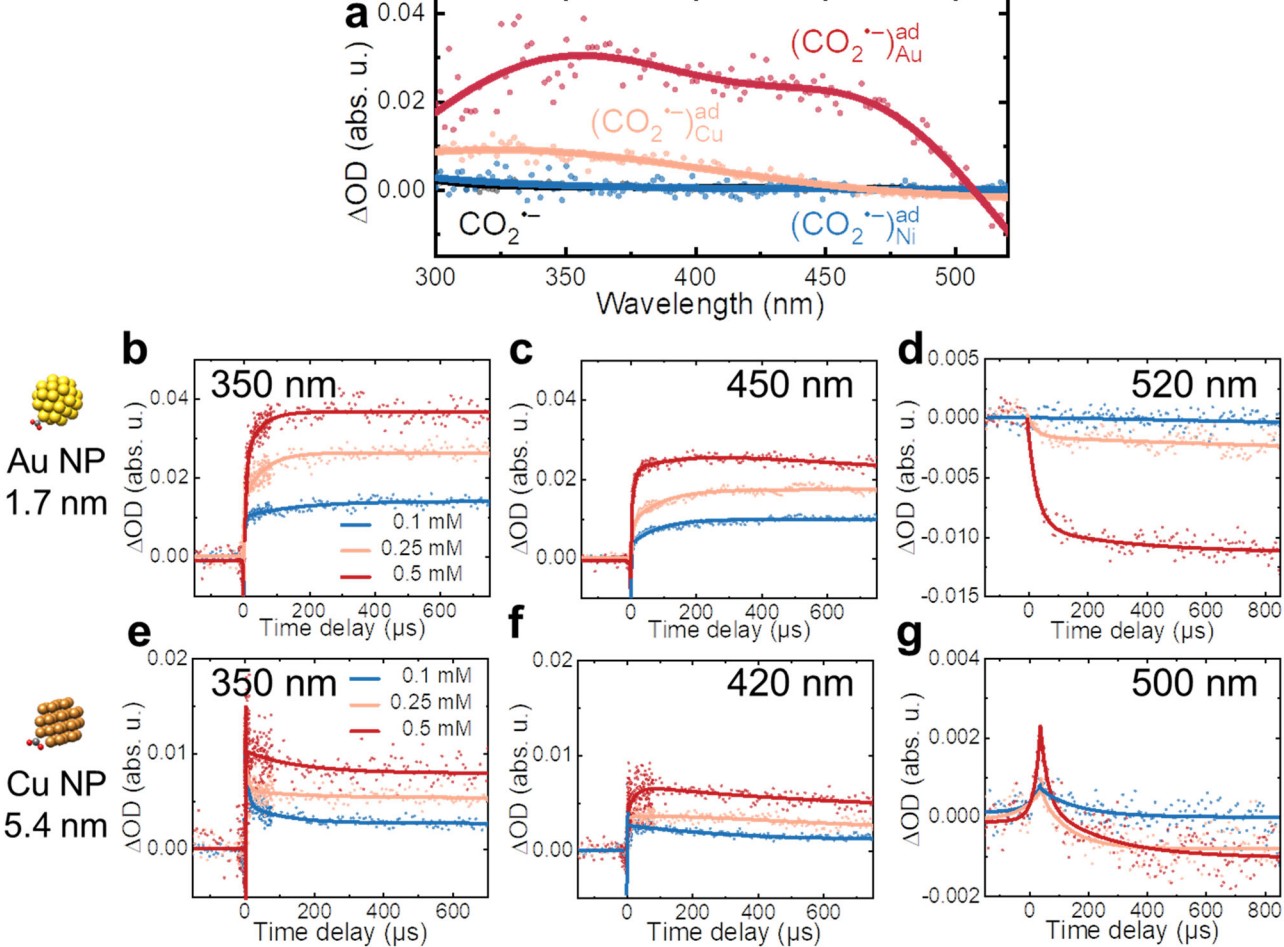

**Fig. 2 | Stability of $(CO_2^{\bullet-})^{ad}$ radicals on metal surfaces.** **a** Transient absorption spectra of $CO_2^{\bullet-}$, $(CO_2^{\bullet-})_{Au}^{ad}$, $(CO_2^{\bullet-})_{Cu}^{ad}$, $(CO_2^{\bullet-})_{Ni}^{ad}$ radicals in 0.5 mM Metal NPs solution at 800 μs. **b–d** Transient kinetics at 350 nm (**b**), 450 nm (**c**), and 520 nm (**d**) with $(CO_2^{\bullet-})_{Au}^{ad}$ as a function of Au concentrations. **e–g** Transient kinetics at 350 nm (**e**), 420 nm (**f**), and 500 nm (**g**) with $(CO_2^{\bullet-})_{Cu}^{ad}$ as a function of Cu concentrations. Source data are provided as a Source Data file.

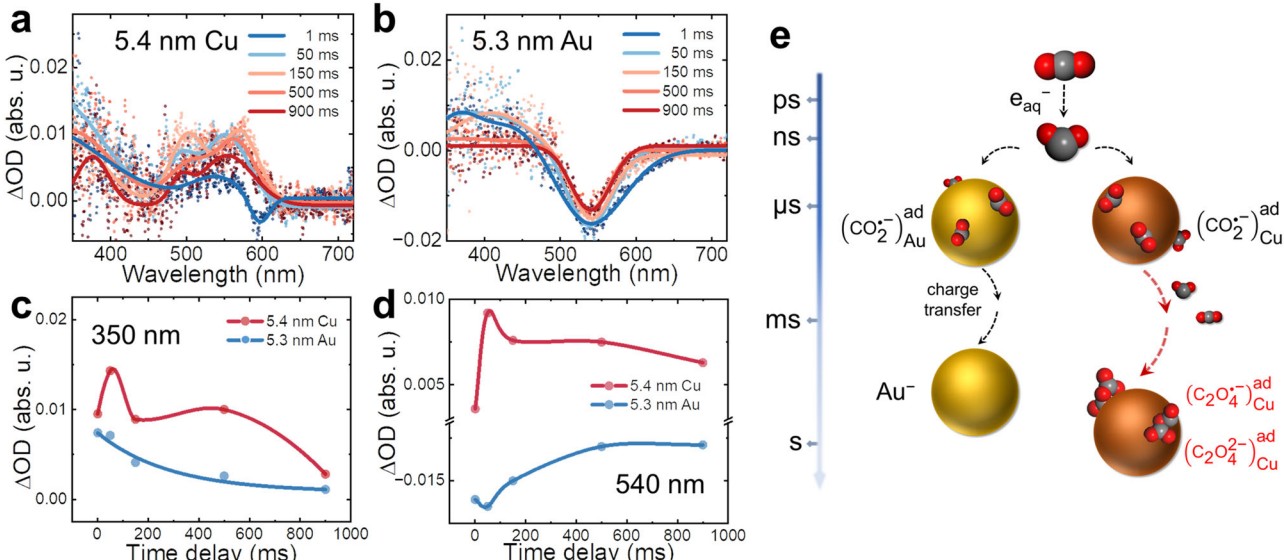

**Fig. 3 | Transient absorption profiles at second timescale. a, b** Transient absorption spectra of $(CO_2^{\bullet-})_{NP}^{ad}$ radicals at 1 ms, 50 ms, 150 ms, 500 ms, and 900 ms in the presence of 0.5 mM 5.4 nm Cu (**a**) and 5.3 nm Au (**b**). **c, d** The curves of ΔO.D. versus time delay at 350 nm (**c**) and 540 nm (**d**). **e** Schematic diagram of the generation, stabilization, and conversion of $CO_2^{\bullet-}$ radicals in the presence of Au and Cu NP. $CO_2$ reacted with $e_{aq}^-$ to form $CO_2^{\bullet-}$ within a few nanoseconds and the $CO_2^{\bullet-}$ radicals stabilized on the surface of Cu and Au NPs. After 1 ms, $(CO_2^{\bullet-})_{Cu}^{ad}$ radicals were converted to dimer intermediates, and $(CO_2^{\bullet-})_{Au}^{ad}$ radicals decayed for around 150 ms. For simplicity, cations are not shown. Source data are provided as a Source Data file.

solutions[10,15,49]. As for Au NPs with similar and smaller sizes, no further reaction occurs; only the decay of the absorption band of the first species is observed, indicating that surface-bound $CO_2^{\bullet-}$ decays slowly within 1 s (Fig. 3b and Supplementary Fig. 10).

Besides, Au and Cu NPs also showed essential distinctions in the kinetics of surface plasmon absorption bands, implying the different interaction between $CO_2^{\bullet-}$ radicals and NPs. Within 1 ms, both Au and Cu present slight bleaching of their surface plasmon absorption band due to the charge of surface-bound $CO_2^{\bullet-}$ radicals, revealing the abovementioned stabilization process. At a longer time (a few hundred ms), once the next reaction happens, a second intermediate species is observed without noticeable bleaching of the absorption band in the case of Cu NPs. Nevertheless, the bleaching of the surface plasmon absorption band in the Au system persisted up to 900 ms, accompanied by the decay of $(CO_2^{\bullet-})_{Au}^{ad}$ radicals, which demonstrated the primary occurrence of charge transfer rather than covalent interaction between Au NPs and $CO_2^{\bullet-}$ radicals (Fig. 3c, d). The different kinetics observations for the bleaching of the surface plasmon absorption band and characteristic surface-bound $CO_2^{\bullet-}$ suggested the different character of Cu and Au for $CO_2^{\bullet-}$ stabilization and conversion, as well as can explain the selective catalysis property of Cu for $CO_2$ reduction.

In order to further validate our findings of $CO_2^{\bullet-}$ coverage, we have performed comprehensive electrochemical $CO_2$RR testing on as-synthesized Au, Cu, and Ni NPs. Our results showed a typical tendency for the model catalysts (Au, Cu, and Ni) in our work (Supplementary Fig. 11). Cu induced the formation of CO, $CH_4$, and $C_2H_4$. In contrast, Ni predominately produced hydrogen with very low activity to convert $CO_2$, and Au mainly produced CO from the conversion of $CO_2$. The tendency of electrochemical $CO_2$RR measurements agrees with our pulse radiolysis observation. We found that Ni NPs cannot stabilize $CO_2^{\bullet-}$ radicals, which correlates with its slight activity of $CO_2$ reduction; however, Cu and Au NPs can stabilize $CO_2^{\bullet-}$ radicals. The long-lived characteristic spectra disclosed that the stabilization process of $CO_2^{\bullet-}$ radicals on the surface of Au and Cu has substantially extended the lifetime of $[CO_2^-]_{ad}$ radicals by at least 100 times compared to $CO_2^{\bullet-}$ radicals in solutions or in the presence of Ni. The extended lifetime lays the groundwork for the subsequent multi-electron transfer reaction. Moreover, the differentiation in the

respective $[CO_2^{\bullet-}]_{ad}$ radical transient kinetics and characteristic spectra across Au, Cu, and Ni systems suggest diverse stabilization behavior and adsorbed structures of $[CO_2^-]_{ad}$ radicals on various metal surfaces. This selective stabilization process determines the subsequent selective reduction pathway of $[CO_2^-]_{ad}$ radicals. Notably, we also observed the dimerization pathway at a subsecond range of surface-bound $CO_2^{\bullet-}$ radicals on Cu rather than Au, which is in accordance with the unique property of Cu to produce $C_2H_4$.

To better understand the nature of distinct absorption bands from $CO_2^{\bullet-}$ radicals absorbed on the surface of Au and Cu NP, molecular simulations were performed at the DFT level with solvent effects represented by a dielectric continuum model. Figure 4a, b depicts the optimized structure of the two complexes and highlights a different interaction mode. Specifically, $CO_2^{\bullet-}$ radical interacts solely via the carbon atom to the Au atom at a distance of 2.17 Å, with geometry comparable to the isolated $CO_2^{\bullet-}$ (see Supplementary Table 1). The overall charge of $CO_2^{\bullet-}$ is −0.5 electrons, indicating a partial electron transfer to Au. A Mayer bond order of 0.45, a covalent interaction provided by the DORI plot, and an Au-C ELF valence basin with a population of 1.2 electrons support this conclusion (see Supplementary Information for computational details). In contrast, $(CO_2^{\bullet-})_{Cu}^{ad}$ stabilizes with the carbon atom and one oxygen atom at distances of 2.01 Å for Cu-C and 2.08 Å for Cu-O. The geometry of $(CO_2^{\bullet-})_{Cu}^{ad}$ is influenced by lowering the COC angle and increasing the intramolecular C-O distances, suggesting a rather strong interaction between $CO_2^{\bullet-}$ and Cu atom. The DORI analysis reveals two covalent interaction sites. Moreover, this larger electron sharing between the two entities is reflected by a Mayer bond order and an ELF valence basin population of 0.85 and 2.5 electrons for Cu-C and 0.54 and 2.2 electrons for Cu-O, respectively. The total spin density of the systems at the ground state was computed and represented in Supplementary Fig. 12. In the case of AuNPs, the spin density of $(CO_2^{\bullet-})_{Au}^{ad}$ remains localized on $CO_2^{\bullet-}$, so it remains a radical species due to a weakly covalent interaction. For Cu, the result highlights a large delocalization of the spin density as a sign of the stronger covalent interaction in $(CO_2^{\bullet-})_{Cu}^{ad}$.

The TD-DFT electronic spectra and the hole-electron framework reveal the global nature of absorption for electronic transitions[50]. The absorption spectra of $(CO_2^{\bullet-})_{Au}^{ad}$ and $(CO_2^{\bullet-})_{Cu}^{ad}$ are found to exhibit a

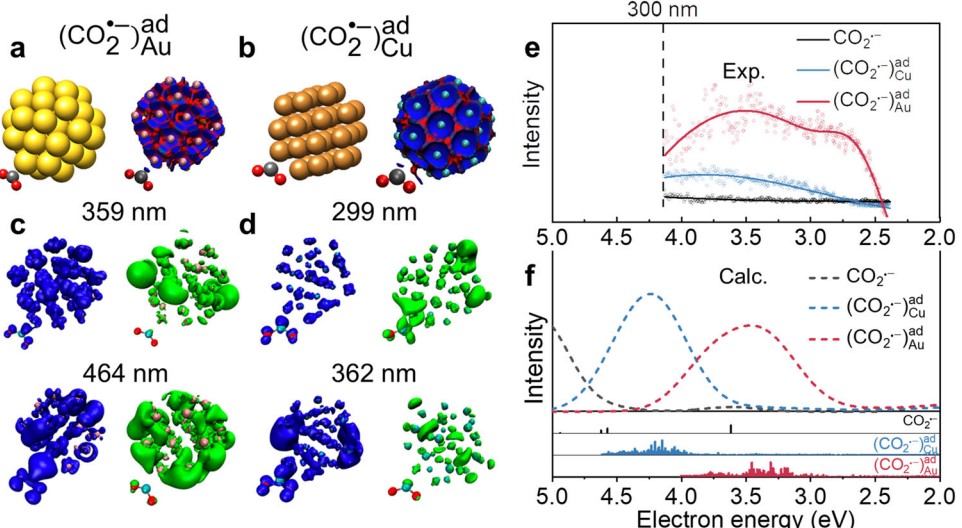

**Fig. 4 | The simulation of CO$_2$$^{•-}$ radical stabilized on metal nanocatalysts.**
**a, b** DFT optimized structures and DORI plots (repulsive forces in red and covalent interaction in blue) of (CO$_2$$^{•-}$)$_{Cu}^{ad}$ (**a**) and (CO$_2$$^{•-}$)$_{Au}^{ad}$ (**b**). **c, d** Electronic transition analysis (hole in blue and electron in green) of (CO$_2$$^{•-}$)$_{Cu}^{ad}$ (**c**) and (CO$_2$$^{•-}$)$_{Au}^{ad}$ (**d**). **e, f** Transient absorption spectra (at 800 μs) (**e**) and TD-DFT electronic spectra (**f**) of CO$_2$$^{•-}$ and CO$_2$$^{•-}$ stabilized on nanocatalysts. Source data are provided as a Source Data file.

red-shift toward isolated CO$_2$$^{•-}$ in solutions, with a band maximum at 358 nm for Au and 293 nm for Cu, in agreement with experimental observation (Fig. 4e). The hole and electron distributions well describe the electron transitions for two representative excited states for each complex in Fig. 4c, d. In the case of Au, the hole distribution is delocalized over NP. It overlaps with CO$_2$$^{•-}$ for the most intense electronic transition at 359 nm, similar at 464 nm with a stronger CO$_2$$^{•-}$ contribution. Compared with reported spectra of CO$_2$ coordinated on Au and Cu surfaces, these surface-bound CO$_2$$^{•-}$ spectra all exhibited red-shift[51–53]. The electron distribution during excitation is mainly delocalized at the surface of NP, reflecting a weak interaction between Au NP and CO$_2$$^{•-}$ radical. However, (CO$_2$$^{•-}$)$_{Cu}^{ad}$ displays a considerable hole distribution of CO$_2$$^{•-}$ radical at 299 and 362 nm, whereas the electron distribution remains localized on several Cu atoms at the surface. The greater localized hole-electron distributions of (CO$_2$$^{•-}$)$_{Cu}^{ad}$ disclose the stronger covalent interaction on Cu surfaces, which is consistent with previous reports[14,54].

### Catalyst size and cation effect on surface-bound CO$_2$$^{•-}$ radical

In catalytic CO$_2$RR systems, two essential factors require thorough consideration. First, the intrinsic activity of nanoscale catalysts depends strongly on the size distribution, surface topology, and surface composition. Second, the complex localized aqueous composition, particularly alkali metal cations, profoundly affects the CO$_2$RR mechanism route through electric double layers and interfacial hydration[55,56]. The (CO$_2$$^{•-}$)$_{Au}^{ad}$ of different sizes Au NP (1.7, 3.3, and 6.2 nm, Supplementary Fig. 13) exhibited a similar wide-range characteristic absorbance from 300 nm to 500 nm at 75 μs, but their intensity increases by reducing the size of nanocatalysts (Fig. 5a). It is evident that when the concentration of Au precursors for the preparation of the nanocatalyst is constant, the smaller size results in a higher concentration of Au NP; consequently, the larger specific surface area provides more active sites. The competition in transient kinetics of CO$_2$$^{•-}$ radicals between dimerization to form oxalate and stabilization on Au surfaces is also size dependent. In the 6.2 nm Au system, the transient kinetics reflect mainly the decay of CO$_2$$^{•-}$ radicals due to the recombination process (Fig. 5b, c). However, as the Au NP size reduces to 1.7 nm, the concentration of nanocatalyst increases almost 27 times, the decay presents an inflection around 7 μs, indicating that the (CO$_2$$^{•-}$)$_{Au}^{ad}$ stabilization mechanism turns to dominate

the competitive processes. As progressed to 750 μs, no apparent changes occur in the spectra shape of (CO$_2$$^{•-}$)$_{Au}^{ad}$ radicals relative to 75 μs (Fig. 5d–f).

More importantly, the increase in Au NP size reduces the intensity ratio between two bands (ΔOD$_{350 nm}$/ΔOD$_{450 nm}$) due to the greater density of surface-bound CO$_2$$^{•-}$ radicals. DFT simulations were performed on the larger size of Au NP with more CO$_2$$^{•-}$ radicals (Supplementary Fig. 14) to investigate this phenomenon. The geometry and orientation of CO$_2$$^{•-}$ radicals on Au surface atoms are the same at interaction sites (the corner position being the most favorable) and similar to the above results for (CO$_2$$^{•-}$)$_{Au}^{ad}$. However, electronic transition analysis indicated the observation of additional states for (CO$_2$$^{•-}$)$_{Au}^{ad}$ at higher wavelengths (especially above 400 nm), where the electrons are more delocalized at the surface of Au NP (Supplementary Fig. 14b). Other simulations on the same size of Au NP with two surface-bound CO$_2$$^{•-}$ reflect that (CO$_2$$^{•-}$)$_{Au}^{ad}$ tends to undergo electron transfer to Au surfaces instead of localized distribution for covalent interaction with Au NP (Supplementary Fig. 15). The aforementioned data show that the smaller Au NP not only provides more active sites for the stabilization reaction but also facilitates localized electron distribution among (CO$_2$$^{•-}$)$_{Au}^{ad}$ radicals and Au surfaces, enhancing the surface-bound interaction.

As reported in the existing studies[57–59], three prominent theories exist regarding the primary function of metal cations during CO$_2$ reduction: modification of the (local) electric field, buffering of the interfacial pH, and the stabilization of intermediates (such as CO$_2$$^{•-}$ radicals) through local field effects. To verify the above assumptions, we carried out experiments in solutions without sodium cations, the samples were centrifugated, and formic acid was used to adjust the concentration of proton. Compared with the above results in the presence of sodium cations, the kinetics of CO$_2$$^{•-}$ radicals stabilizing on Au surfaces was retarded, and the absorption intensity of surface-bound CO$_2$$^{•-}$ radicals on Au NPs remarkably decreased (Supplementary Fig. 16a–d), indicating that the stabilization process of CO$_2$$^{•-}$ radicals on Au surfaces was undermined in the absence of metal cations. Moreover, transient spectro-kinetics data revealed that the lifetime of surface-bound intermediates without metal cations was curtailed regarding the sodium system (Supplementary Fig. 16e–g), suggesting that the primary role of cations is to stabilize the surface-bound CO$_2$$^{•-}$ radicals. These observations are in total agreement with recent steady-

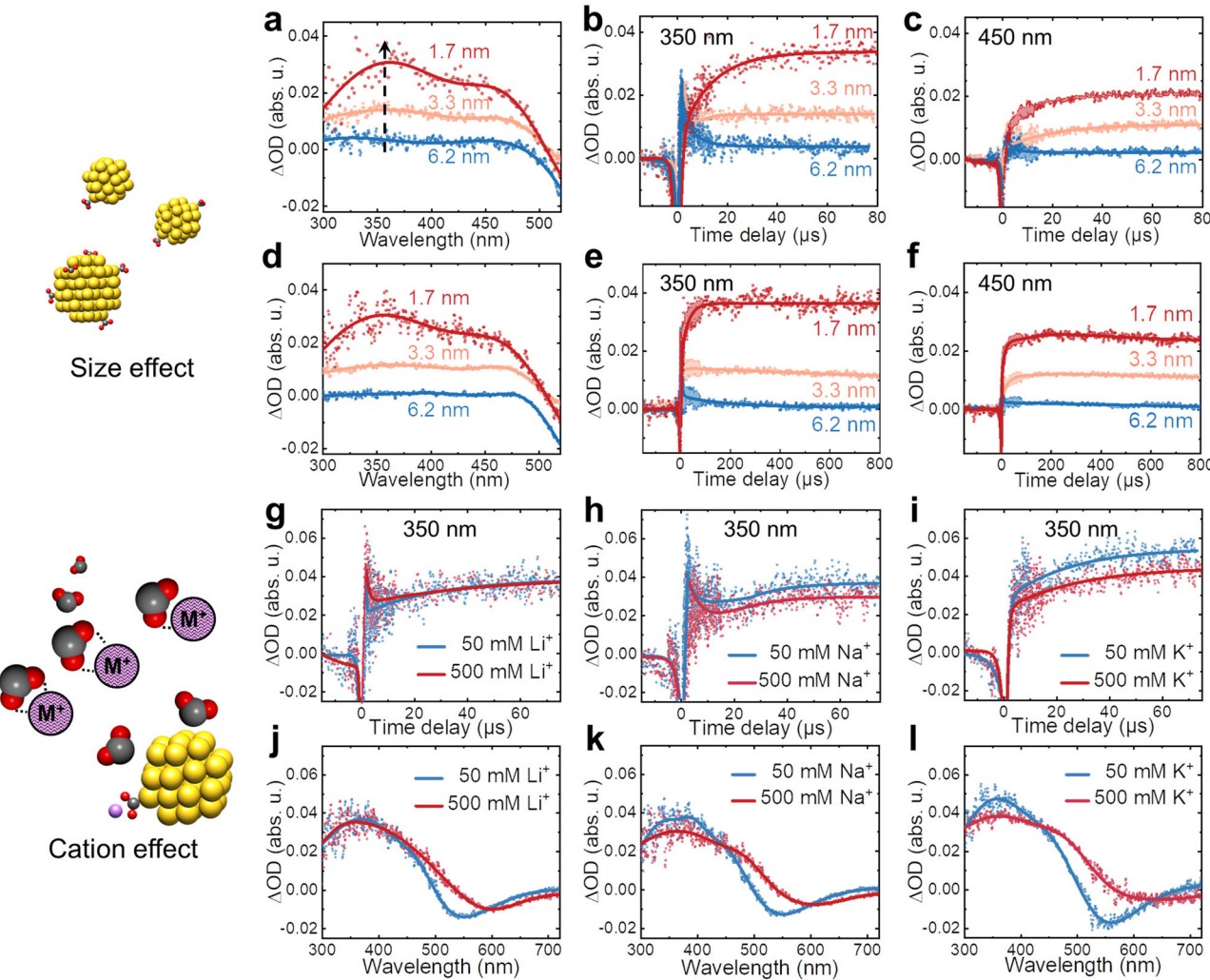

**Fig. 5 | The effect of catalyst size and cation in solutions on $CO_2^{\cdot-}$ radical stabilization process with Au. a–c** Transient absorption spectra of different sizes of Au (0.5 mM) at 75 μs (**a**) and transient kinetics at 350 nm (**b**) and 450 nm (**c**) within 80 μs. **d–f** Transient absorption spectra of different sizes of Au (0.5 mM) at 750 μs (**d**) and transient kinetics at 350 nm (**e**) and 450 nm (**f**) within 800 μs. **g–i** Transient kinetics at 350 nm with different concentrations of Li$^+$ (**g**), Na$^+$ (**h**), K$^+$ (**i**). **j–l** Transient absorption spectra at 750 μs with varying concentrations of Li$^+$ (**j**), Na$^+$ (**k**), K$^+$ (**l**). Source data are provided as a Source Data file.

state electrocatalysis studies[57–59]. However, excess metal cations disturb the stabilization of intermediates through electric and nonelectric field aspects[60–63]. Figure 5g shows that adding Li$^+$ does not affect the transient kinetics of $CO_2^{\cdot-}$ at 350 nm. The same transient kinetics rule out the possible interference of Na$^+$ from HCOONa during radiolytic preparation when replacing HCOONa with HCOOLi for radiolytic preparation (Supplementary Fig. 17a, b). The simulation of the complex $(CO_2^{\cdot-})_{Au}^{ad}$ with Li$^+$ also led to a stable structure with an electrostatic interaction between cation and $CO_2^{\cdot-}$ while keeping a similar orientation with Au. While Fig. 5h, i indicates that elevated concentrations of both Na$^+$ and K$^+$ suppress the increase of transient absorption within the first 5–10 μs, after this period, the transient kinetics exhibits no dependence on cation concentration until 750 μs (Supplementary Fig. 17c–e). These results demonstrate that metal cations primarily interfere with the diffusion and initial stabilization process of $CO_2^{\cdot-}$ radicals on Au surfaces. According to previous reports[61,63], we infer that metal cations combine with $CO_2^{\cdot-}$ radicals through electrostatic interactions to form ion pairs ($M^{z+} + CO_2^{\cdot-} \leftrightarrow (M^{z+} \bullet\bullet\bullet CO_2^{\cdot-})$) that obstruct the stabilization of $CO_2^{\cdot-}$ radicals on Au surfaces. Also, the trend in transient absorption spectra intensity follows the order Li$^+$ < Na$^+$ < K$^+$ for the same concentration (Fig. 5j–l), which correlates with the cation radius and supports the hypothesis of the pair formation. In contrast to the reported steady-state results, our finding fills in the missing

information on the role of metal cations during the stabilization process of $CO_2^{\cdot-}$ radicals.

In summary, pulse radiolysis allows us to acquire a complete image of the stabilization process of $CO_2^{\cdot-}$ radicals on catalyst surfaces. Even if the pathway to form surface-bound $CO_2^{\cdot-}$ radical on the NPs is fundamentally different between electrocatalysis and radiolysis conditions, the final state ($CO_2^{\cdot-}$ bound on the NPs) is the same as whatever is produced by radiolysis or direct reduction of $CO_2$ on the surface of the electrode. The surface stabilization process of $CO_2^{\cdot-}$ radicals on Au and Cu surfaces includes the initial adsorption and the structure optimization process, which extend substantially the lifetime of $(CO_2^{\cdot-})^{ad}$ radicals by at least 100 times compared to $CO_2^{\cdot-}$ radicals in solutions, thereby laying the groundwork for the subsequent multielectron transfer reaction. Most notably, the respective $(CO_2^{\cdot-})^{ad}$ radical transient kinetics and characteristic spectra across Au, Cu, and Ni systems suggest diverse stabilization behavior and structures of $(CO_2^{\cdot-})^{ad}$ radicals on various metal surfaces. Extensive research in electrochemistry has resulted in categorizing metal electrodes into four distinct groups based on their primary products. The different selectivity of $CO_2$ reduction between metals has been explained by their binding energy to key intermediates, including *OCHO, *COOH, and *CO. Therefore, we assume that the stabilized structure and kinetics of the first key intermediate, surface-bound $CO_2^{\cdot-}$ on metallic sites, influence

the catalytic properties and the product selectivity. Au, Ni, and Cu NPs distinctly represent the individual catalytic selectivity and activity of each group. We observed very clearly the difference between the three nanocatalysts. In the same conditions, we found that Ni cannot stabilize the $CO_2^{\bullet-}$ radicals, which corroborates its slight activity of $CO_2$ reduction reported in the literature; however, Au and Cu can stabilize the $CO_2^{\bullet-}$ radicals weakly and strongly, respectively, in agreement with the results obtained by steady-state electrocatalysis. The importance of our work lies not only in explaining the reaction mechanism in electrocatalysis but also in deciphering the function of the metal in $CO_2$ reduction. This selective stabilization mechanism provides deeper insight into the intrinsic mechanism of $CO_2$ reduction, extending from steady-state intermediates to transient elementary reactions. Our results are a significant advance in the development of time-resolved techniques for optimizing catalyst design and system performance.

## Methods

### Chemicals

Cetyltrimethylammonium chloride (CTAC, 98%), sodium formate (99%), copper(II) sulfate (98), nickel(II) chloride (98%), sodium tetrachloroaurate (99%), and cetyltrimethylammonium bromide (98%) were purchased from Sigma-Aldrich. All chemicals were used without further purification. Ultrapure water with a resistivity of 18.25 MΩ cm was obtained from a water purification system. $CO_2$ (99.999%) and Ar (99.999%) were purchased from Air Liquide Industrial Gases Company.

### Synthesis of metal nanoparticles

All metal nanoparticles were synthesized by radiolytic methods: In a typical synthesis for Cu nanoparticles, 20 mmol sodium formate and 0.5 mmol CTAC were added to 10 mL $H_2O$. Then, 10 mL copper(II) sulfate (1 mmol $L^{-1}$) solution was added to form homogeneous solutions. The solutions were saturated with Ar and sealed in a homemade cuvette. The cuvette was irradiated at a $^{60}Co$ gamma source (5.35 × $10^{13}$ Bq, located in the Université Paris-Saclay) at a dose rate of 2.2 kGy $h^{-1}$ at ambient conditions. The dose rate was calibrated using the Fricke dosimeter, and the total absorbed dose reached 6.6 kGy. The resulting Cu nanoparticle solution was directly used for pulse radiolysis experiments and absorption spectroscopy characterization.

For Ni or Au nanoparticles, the precursor was replaced with nickel(II) chloride or sodium tetrachloroaurate. For size modification, it was controlled by varying the concentration of CTAB, with higher concentrations resulting in larger particles. The synthesized 1.7, 5.3, and 6.2 nm Au nanoparticles correspond to 1, 2.5, and 10 mM CTAB concentration systems. The supernatant solution was centrifuged at 164,700×$g$ for 24 h using an ultracentrifuge (Optima XE-90, BECKMAN COULTER) to remove the NPs.

### Characterization

Particle morphology and size distribution were examined by high-resolution transmission electron microscopy (HR-TEM, JEOL 2100 PLUS, 150 kV). UV-vis spectra were recorded on a UV-spectrophotometer (HEWLETT PACKARD 8453, HP) in the range from 190 to 800 nm.

### Pulse radiolysis

Pulse radiolysis experiments were carried out employing the picosecond laser-triggered electron accelerator, ELYSE, coupled with a time-resolved absorption spectrophotometric detection system[39]. Laser (260 nm) driven Cs₂Te photocathode allowed the production of short electron pulses with a typical half width of 7 ps, a charge of ≈6 nC, and energy of ≈7.8 MeV at a repetition rate of 10 Hz. During irradiation, metal NP solutions were contained in a homemade cell with a path length of 1 cm. The diameter of the electron beam was 3 mm, and the irradiated volume was less than 0.1 mL.

Absorption spectral measurements were performed using the white light from a homemade Xenon flash lamp. The light was focused on the sample parallel to the electron beam with a smaller diameter and then directed onto a flat field spectrograph (250IS, Chromex), which disperses the light on the entrance optics of a high dynamic range streak-camera (C-7700-01, HAMAMATSU) to obtain an image resolved in wavelength and time. The kinetic data and absorption spectra were extracted from three series of 250 resulting images. In this work, the transient spectra were measured from 290 to 720 nm at 1 μs, 10 μs, 20 μs, 100 μs, and 1 ms[40].

### Computational methods

DFT calculations were done using the ORCA 5.0.3 software[64]. The geometry optimization calculations were performed using the PBE functional, the SDD pseudopotential (with 19 valence-electrons treated explicitly) for the metal atoms, and the def2-TZVP basis set for carbon and oxygen atoms. Dispersion corrections were added to the functional used in the D3 framework proposed by Grimme with the addition of the Becke–Johnson damping (D3BJ) in all cases[65]. $(CO_2^{\bullet-})_M^{ad}$ (M for Au and Cu) has been designed as a model of a metallic NP in interaction with one $CO_2^{\bullet-}$ radical with solvent effects represented by a dielectric continuum model. The orientation and nature of the interaction between the radical and NP have been probed by several quantum chemical analyses. Numerical frequencies were calculated to ensure the structures corresponded to energy minima. For Au and Cu, the 38-atom nanoparticle is considered in the singlet spin state, and the complex with the radical results in a doublet spin state. The CPCM implicit solvent model was used to represent the water environment. The electronic spectra were computed in the TD-DFT framework using the CAM-B3LYP functional and the implicit solvent. Between 600 and 800 excited states were included in the calculations to obtain the spectra of the different metal systems. The TD-DFT spectrum of the isolated $CO_2^{\bullet-}$ radical was obtained at the CAM-B3LYP/aug-cc-pVTZ level by calculating 200 excited states. The structure of the 92-atom Au NP with 5 $CO_2^{\bullet-}$ radicals was optimized at PBE-D3BJ level (with the SVP basis set for C and O atoms) using an implicit solvent. A simplified calculation of the electronic spectrum was computed in vacuum using the simplified sTDA[66] approach and the wB97X functional. The quantum chemical topological and electronic transition analyses were performed with the Multiwfn code by analyzing the electron density generated from ORCA[67]. The analyses of both covalent and non-covalent interactions were calculated based on the Density Overlap Regions Indicator (DORI)[68]. The electron localization function (ELF) was calculated to determine the valence basins[69].

### Reporting summary

Further information on research design is available in the Nature Portfolio Reporting Summary linked to this article.

## Data availability

The data that support the findings of this study are available via Zenodo[70] and from the corresponding authors upon request. Source data are provided with this paper.

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

## Acknowledgements

This work was supported by the National Natural Science Foundation of China (11975122, 22006067, and 21906083), Jiangsu Province Science Fund for Distinguished Young Scholars (BK20230032), Fundamental Research Funds for the Central Universities (No. NE2020006), Scientific and Technological Innovation Special Fund for Carbon Peak and Carbon Neutrality of Jiangsu Province (BK20220026), and Jiangsu Funding Program for Excellent Postdoctoral Talent (2022ZB197). We thank Jean-Philippe Larbre for his help during the experiments at ELYSE.

## Author contributions

Z.W.J., C.C., J.M., and M.M. conceived of the work; Z.W.J. and M.M. designed and performed the experiments; S.A.D. performed pulse radiolysis experiments on the second timescale; C.C. performed the DFT calculations; S.N.S. and F.H. performed the Tafel analysis and electrochemical catalysis measurements. Z.W.J., C.J.H., J.M., and M.M. analyzed the data; Z.W.J. and C.C. wrote the manuscript; C.C., J.M., and M.M. deeply revised the manuscript.

## Competing interests

The authors declare no competing interests.
