## [Peer Review File · Nature Communications]

Direct Time-resolved Observation of Surface-Bound Carbon Dioxide Radical Anions on Metallic NanocatalystsREVIEWER COMMENTS

Reviewer #1 (Remarks to the Author):

In this manuscript, the authors utilized pulse radiolysis to observe the holistic stabilization process of $\text{CO}_2^{\bullet-}$ radicals under working CO_2RR conditions on well-defined nanoscale metallic sites, which allows the identification of surface-bound intermediates with characteristic transient absorption and the distinct kinetics on three typical metallic nanocatalysts (Cu, Au, and Ni). The combined spectra evidence with molecular simulations provide insightful understanding of the selectivity of the critical initial step of CO_2RR . Overall, this work realizes time-resolved investigation on the stabilization process of $\text{CO}_2^{\bullet-}$ radicals during reactions and offers unprecedented understanding of CO_2RR mechanism. I suggest major revisions are quite required before publication.

My concerns as follows:

1. Firstly, main conclusions obtained in this work are strongly dependent on the utilization of pulse radiolysis for providing time-resolved spectra, however, the time resolution of this technique is not provided. Moreover, the authors only offered a one-sentence discussion on the advantage of pulse radiolysis as compared with those spectroscopic techniques based on molecular vibration or photoelectron excitation, it is hard to understand the fundamentals and powerful capabilities of this method in achieving desired information.
2. In this work, the authors claimed that the initial transient elementary reaction during CO_2RR is the critical stabilization process and diverse stabilization behaviors on different metallic sites. How is the correlation between the observed stabilization behaviors and the CO_2RR activity/selectivity? The lacking understanding in this aspect is the weak of this work.
3. The reviewer wonders whether the stabilization process of $\text{CO}_2^{\bullet-}$ radicals is the RDS for the CO_2RR . Although the authors mentioned in introduction that the existence of $\text{CO}_2^{\bullet-}$ radicals on metal electrodes, such as Au, Cu, and Ag, has been proposed based on the experimental Tafel slope analysis, experimental evidence and related discussions are missing in this work.
4. Can the time-resolved measurements be performed at applied potentials related to CO_2RR . In this way, the time-resolved investigations is more meaningful for understanding the initial step for CO_2RR .
5. In Fig. 2, the authors discussed the effect of metallic sites on the stabilization process of $\text{CO}_2^{\bullet-}$ radicals. However, the surface chemical states are not considered since Cu and Ni are prone to be oxidized under current experimental conditions. Thus, the effect of surface oxidized species on surface-bound $\text{CO}_2^{\bullet-}$ radicals should be also included in the last section.
6. For Fig. 3e, it is strongly suggested a direct comparison between the experimental data with the calculation results, as well as data analysis and discussion.
7. There are so many typos and grammar mistakes in the manuscript. For instance, line 206, page 11. The authors should carefully check them throughout the manuscript.

Reviewer #2 (Remarks to the Author):

I am not a specialist in radiolysis but the experiment described in this paper is interesting and valuable. However, there are some major revisions needed.

The authors refer extensively to electrochemistry and electrocatalysis, but there is no electrochemistry or catalysis (or catalytic reaction) in their experiment, as far as I can see. The authors need to provide a more detailed description of how exactly their experiment relates to electrocatalysis (for instance, the issue of electrode potential). It is believed that under electrochemical conditions, the CO_2 radical cation is only formed at very negative potentials. How does that relate to the authors' experiments? Another important is the role of cations. The authors seems to have missed the recent discussions on the role of cations in CO_2 reduction, some literature claiming that cations are indispensable for CO_2 reduction. It is a pity that this paper does not relate to that discussion. In particular, an experiment with no cations (only protons) would be have been potentially extremely insightful. Entries into the

literature: <https://www.nature.com/articles/s41929-021-00655-5>,
<https://pubs.acs.org/doi/10.1021/jacs.2c11643>

In the abstract, what is meant with the word "holistic"?

In the introduction, please avoid presenting CO₂ reduction as a way to mitigate CO₂ emission. The only way to mitigate CO₂ emissions is to not emit it.

Reviewer #3 (Remarks to the Author):

This work contributes to the efforts to understand the CO₂ electroreduction reaction better. I find this work very interesting. Its strong part is the use of a time-resolved technique, an interesting objective (transient kinetics of the coverage of the CO₂•⁻ radical adsorption, the coverage of the important effects of the catalyst metal, the nanoparticle size (NP), and the electrolyte cation, as well as the use of DFT to interpret their experimental data. However, I recommend the authors to address the following minor comments to make the manuscript publishable.

1. Line 40 The statement "The existence of CO₂•⁻ on metal electrodes, such as Au, Cu, and Ag, has been proposed based on the experimental Tafel slope analysis." Needs a reference
2. Improve grammar/style: For example: change "free nanocatalyst solution" to "nanocatalyst-free solution"
3. Fig. 1 in the main text and all similar figures have unclear/incomplete captions: For example, the caption of Fig. 1 should mention that (a) is Transient kinetics at 350 nm within 700 ns "as a function of metal concentration", (b) and (d) should specify the metal concentration. The same is applicable to the analogous graphs
4. Lines 131-133: "The slower kinetics observed in the Cu system than Au can be attributed to its lower affinity towards CO₂•⁻ radicals or to the lower NP concentration resulting from the larger size of Cu (5.41 nm) than Au (1.71 nm)." This statement is questionable, as Cu is stronger interacts with CO₂ than Au doi:10.1021/jp074570y. The authors also have their own computational simulation results to verify whether this is true or not.
5. Line 135 "...different kinetics of [CO₂•⁻] between Au and Cu suggests that the distinct stabilization reaction and structure may result in selective catalysis." This conclusion does not have any ground unless the authors normalize the reaction rate by the surface areas of NPs. It is unclear why the comparison in the main text is done for the NPs with different size, even though the authors have the same sizes (5.4 Cu and 5.3 Au nm). The same argument applies to the discussion of the NC size effect on the reaction kinetics on p.11
6. Line 151 "...transient kinetics to the millisecond range" Should be changed to "submillisecond range"
7. Line 161 "plasma band" should be changed to "electron plasma band"
8. On p. 10, the authors discuss the difference in the coordination of the CO₂•⁻ radical with Au and Cu clusters. The comparison with the lit data on the coordination of CO₂ on the bulk surfaces would be recommended
9. Lines 238-239: the recommended citations relevant to the effect of cation on the interfacial reactivity of CO₂ are doi:10.1039/C8CP07807F and doi:10.1038/s41929-021-00655-5
10. How the spin state of the radical changes upon adsorption? Is it still radical?

Reviewer #4 (Remarks to the Author):

In this work, the authors performed time-resolved visible spectroscopy of CO₂/metal nanoparticle systems, following a radiolysis pulse that induced CO₂*⁻ radical formation. The authors claim that they observed the transient behavior of the surface-bound CO₂*⁻ radical, which is thought to play a key role in electrocatalytic CO₂ reduction. They explored the kinetics as a function of nanoparticle size

and electrolyte conditions. Overall, this work appears technically solid. The manuscript is mostly well-written. However, I do have significant concerns in regards to the interpretation of the data and the transferability of the insights to catalytic systems:

(1) It is unclear if the observed transients attributed to surface-bound CO_2^* are simply due to electron transfer to the nanoparticles (NPs). Au and Cu NPs exhibit significant absorbances at wavelengths below 600 nm. Therefore, electron transfer to these NPs from a species other than CO_2^* could give rise to the observed transient absorbances and altered kinetics (e.g., Fig. 1b). The authors should conduct a control experiment with a system that contains all components but CO_2 . This is a critically important control experiment; in the absence of such an experiment, their transients cannot be conclusively attributed to CO_2^* -NP adducts.

(2) Assuming that the authors can confirm that the transients arise from CO_2^* -NP adducts through appropriate control experiments, it is unclear to me how the reported information can be transferred to catalytic systems: (a) In an electro- or photocatalytic system, CO_2 adsorbs on the surface. Electron transfer to CO_2 occurs either concurrent with the adsorption or follows initial physisorption. In the system investigated by the authors, CO_2^* is first prepared in solution, which then adsorbs on the NP. So, the pathway is fundamentally altered. This strongly limits the transferability of the results to actual catalytic systems. For example, the authors investigated cation effects on the CO_2^* - kinetics. However, they found that CO_2^* forms adducts with cations in solution, before CO_2^* adsorption. Obviously, this pathway is not operational in a catalytic system. For this reason, I question the broader impact of the study. (b) Once CO_2^* has formed on the surface, the technique can – in principle – characterize its decay kinetics. However, it is unclear to me what the extracted kinetics mean in a broader context. That a metallic surface stabilizes the CO_2^* radical is not surprising. I assume that statistically, only one e^-/CO_2^* reaches a given NP. So the reaction does not further proceed. A more impactful experiment would follow-up with a second radiolysis pulse to convert the surface-adsorbed CO_2^* -radical to CO. The conversion kinetics of this step would be important to characterize, but it is not done in this study.

(3) The authors consistently refer to the intermediate at CO_2^* . They determined that it persists on timescales of 100s of microseconds. In an aqueous solution, it appears unlikely that the radical persists on such long timescales. Although it is generally accepted that e^- transfer precedes H^+ transfer for this first step of CO_2 reduction, protonation of the radical is expected to be very fast. The authors should discuss this possibility in the manuscript. In this regard, the language of the stabilization process is diffuse. For example, on p. 7, the authors note that “the coordination state progressively orientates and sustains for more than 80 microseconds”. It is unclear what this stabilization process entails.

(4) On p. 7, the authors note that “the slower kinetics observed in the Cu system than Au can be attributed to its lower affinity towards CO_2^* radicals”. Though this is suggested as one of two possibilities, it appears an unlikely one. I would expect Au to be generally less reactive than Cu (few things adsorb on Au strongly). The authors statement should be properly qualified. Further, the authors may want to acknowledge other possibilities. For example, is it possible that the CO_2^* radical on Cu decays to another species faster than on Au?

(5) The sentences on p. 9 are confusing. The authors first discuss the “plasma band”, then they refer to the same band as the “plasmon band”. This should be corrected. This is also the only time in the manuscript that they use this terminology. It would be better to introduce this term on p. 4 when discussing Figure S1 (or avoid it altogether). I also recommend that in the sentence “We suggest that the rise of the bleaching signal with Au content, ...”, Figure 2g should be referenced.

(6) The actual data in the figures are difficult to see. Typically, only the fit to the data can be clearly seen. The authors should show the experimental data points more clearly (use more intense colors). Also, it is unclear in Fig. 1b, f, and j if the data shown in these 3D figures represent a fit to the data or the actual data (same for Figure S1). This should be clarified. Judging from some of the SI figures, it appears that the spectra are much noisier than shown in these graphs. I also note that while the 3D perspective view looks nice, it makes it impossible to compare the amplitudes across spectra.

(7) There are some minor language problems throughout the manuscript. I recommend that the authors thoroughly read it and correct any mistakes. Examples: The plural “s” is missing in “nanocatalysts” in the title; p. 5: “we performed the control experiments” should read “we performed

control experiments"; p. 10: "simulations are performed" should read "simulations were" performed.

Point-by-point responses to the reviewers' comments

(Manuscript ID: NCOMMS-23-18719-T)

We have thoroughly analyzed the feedback provided by the reviewers and made significant improvements to our manuscript accordingly. The revised version now encompasses a broader range of experimental and theoretical findings, resulting in a more comprehensive study with enhanced clarity and references. We conducted four additional studies to strengthen our conclusions:

1. Control experiments: To validate our results, we performed an experiment without CO₂, now included in the manuscript, along with a new figure in the Supplementary Information (SI). This experiment serves as an important control to confirm our observations.
 - I. Sample without nanoparticles (NPs) - No absorption band is observed.
 - II. Sample without CO₂ - No absorption band is observed.
 - III. Effect of NP concentration – For a given CO₂^{•-} concentration (which is fixed by dose per pulse), the intensity of the absorption band depends on this scavenging rate of the radical, thus varying with NP concentration.
 - IV. Effect of NP size - We observed that for a given atomic concentration, larger NPs exhibit the lower intensity of the absorption band due to their lower NP concentration and reduced efficiency in CO₂^{•-} radical scavenging.

We have achieved a significant milestone in our research by conducting four control experiments that provide robust support for our conclusions regarding the attribution of the absorption band. These control experiments serve as a crucial foundation, reinforcing the validity and reliability of our findings.

2. The effect of the cation: To address the role of the metal cation, we conducted additional experiments. By preparing metal cation-free samples, we successfully observed the interaction between the radical and the NPs, establishing the significance of the metal cation in our study. Our observations show that the metal cation favors the interaction between CO₂^{•-} radical and NPs. This observation agrees with the recent indirect observations published in Monteiro, Mariana CO, et al. *Nature Catalysis* 4, 654-662 (2021).
3. Prolonged observation of CO₂^{•-} activity on NP surfaces: We have extended the duration of our observations from 1 ms to one second, as suggested by the reviewer. The results are remarkable! Our observations now reveal a distinct disparity in the behavior of the radical's interaction with Cu and Au NPs. In the case of Cu NPs, we observed a new intermediate with characteristic absorption bands, whereas with Au NPs, we observed only a transient decline without any additional features.
4. Spin state of the CO₂^{•-} radical on the NP surface: In response to a question raised by the reviewer, we conducted new numerical simulations to explore the spin value at the ground state when the radical is on the NP surface. An intriguing finding emerged: the overall spin of the radical on gold NPs was determined to be 1/2, as the interaction between the radical and the gold NP was relatively weak. Conversely, for copper NPs, a genuine bond exists between the radical and the copper, leading to electron distribution onto the copper NP. Consequently, the species ceases to be radicals.

We have also responded one by one to all the comments made by the reviewers (please see the below file), and we are sincerely grateful to the reviewers for their comments.

Reviewer #1:

In this manuscript, the authors utilized pulse radiolysis to observe the holistic stabilization process of $\text{CO}_2^{\bullet-}$ radicals under working CO_2RR conditions on well-defined nanoscale metallic sites, which allows the identification of surface-bound intermediates with characteristic transient absorption and the distinct kinetics on three typical metallic nanocatalysts (Cu, Au, and Ni). The combined spectra evidence with molecular simulations provide insightful understanding of the selectivity of the critical initial step of CO_2RR . Overall, this work realizes time-resolved investigation on the stabilization process of $\text{CO}_2^{\bullet-}$ radicals during reactions and offers unprecedented understanding of CO_2RR mechanism. I suggest major revisions are quite required before publication.

Response: We thank the reviewer for the positive comment on the unprecedented understanding of the CO_2RR mechanism brought by the manuscript.

My concerns as follows:

1. Firstly, main conclusions obtained in this work are strongly dependent on the utilization of pulse radiolysis for providing time-resolved spectra. However, the time resolution of this technique is not provided. Moreover, the authors only offered a one-sentence discussion on the advantage of pulse radiolysis as compared with those spectroscopic techniques based on molecular vibration or photoelectron excitation; it is hard to understand the fundamentals and powerful capabilities of this method in achieving desired information.

Response: Thanks for the reviewer's reminder. The time resolution of our pulse radiolysis techniques is 7 picoseconds (ps) which is very short. That means the hydrated electrons which form $\text{CO}_2^{\bullet-}$ radicals in bulk are produced within 7 ps. However, the rates of the interfacial reactions involved in this study are near diffusion controlled. Therefore, our time resolution is much shorter than what we need.

The hydrated electrons first diffuse to react with CO_2 forming $\text{CO}_2^{\bullet-}$ radicals. The rate constant of this reaction is well-known ($8 \times 10^9 \text{ M}^{-1} \text{ s}^{-1}$), and in our condition, $\text{CO}_2^{\bullet-}$ radicals are formed within 10 nanoseconds (ns). Subsequently, $\text{CO}_2^{\bullet-}$ radicals could react with the NPs. The rapidity of this reaction depends on the available surfaces (imposed by the concentration and size of NPs). By probing the spectral evolutions, the reaction to establish a bond between $\text{CO}_2^{\bullet-}$ radical and NPs occurs within a few tens of microseconds (μs). The lifetime of surface-bound $\text{CO}_2^{\bullet-}$ radicals is about a few milliseconds and can be transformed into another surface intermediate in the second range. We now successfully observed all these reactions by the picosecond pulse radiolysis method, and the important finding is that $\text{CO}_2^{\bullet-}$ when it is on the surface of NPs exhibits different electron distribution and transient kinetics. This could provide fundamental insights into the electrocatalytic, photocatalytic, and emerging radiation-driven catalytic CO_2RR .

Until now, the current in situ/operando spectroscopic studies on CO_2 reduction were performed with subsecond time resolution mainly based on infrared, Raman, MS, and XAS (Ren, H. *et al. Nat. Catal.* **5**, 1169–1179 (2022).; Timoshenko, J. *et al. Nat. Catal.* **5**, 259–267 (2022).; Sheng, H. *et al. J. Am. Chem. Soc.* **140**, 4363–4371 (2018).; Zhang, H. *et al. Nat. Commun.* **13**, 6029 (2022).). Our radiolysis setup, coupled with transient absorption spectra, allows for the direct and quantitative observation of the fast reaction between $\text{CO}_2^{\bullet-}$ and well-defined nanoparticles. By extending the delay time, the transformation kinetics of surface-bound $\text{CO}_2^{\bullet-}$ that takes place from milliseconds to seconds can also be investigated. More importantly, the radiolysis method initially generates reactive species in bulk solutions, thus avoiding potential supported-catalysts perturbation and the complexity often encountered with photosensitizers or electrodes. Therefore, pulse radiolysis allows us to focus on the first elementary step of CO_2 reduction, the binding on the nanoscale metallic sites, and its stability on the surfaces, which greatly excludes the solution effects, competing reactions, and surface structure evolution or oxidation of catalysts.

We supplemented these descriptions in the revised manuscript as proposed by the reviewer. Please find them on P3 to P4.

2. In this work, the authors claimed that the initial transient elementary reaction during CO₂RR is the critical stabilization process and diverse stabilization behaviors on different metallic sites. How is the correlation between the observed stabilization behaviors and the CO₂RR activity/selectivity? The lacking understanding in this aspect is the weak of this work.

Response: Extensive electrochemistry studies have established the dependence of CO₂RR products on the categorization of metal electrodes (Wang, G. *et al. Chem. Soc. Rev.* **50**, 4993-5061 (2021).; Nitopi, S. *et al. Chem. Rev.* **119** 7610-7672 (2019).). The different selectivity of CO₂ reduction between metals has been attributed to their binding energy to key intermediates, including *OCHO, *COOH, and *CO, mostly by theoretical calculations. However, the timescale of the initial reaction and exact spin states of the intermediates have been the missing point. Therefore, we assume that the electronic structure, as indicated by absorption spectra, and kinetics of the first key intermediate, surface-bound CO₂^{•-} on metallic sites, can be interesting experimental observables that readily correlate with the established theory of CO₂RR activity/selectivity. Therefore, we attempted three distinct catalytic sites, Au, Ni, and Cu, to fill the gap and validate our assumption.

We observed very clearly the difference between the three systems. In the same conditions, we found that Ni NPs cannot stabilize CO₂^{•-} radicals, which corroborates its slight activity of CO₂ reduction reported in the literature; however, Cu and Au NPs can stabilize CO₂^{•-} radicals, and the binding intermediates exhibit a specific absorption band as well as longer (100 times) lifetime than CO₂^{•-} radicals without NPs. In the case of Cu, we observe a faster decay of bound CO₂^{•-} radicals than Au. We observed that CO₂^{•-} radicals on Cu decay to form another species. These kinetics data show that Cu and Au are both effective catalysts for CO₂ reduction, but Cu NPs are more efficient. According to our results, the selectivity of CO₂ reduction may be predetermined during the binding step of CO₂^{•-} radicals, well before the binding step of intermediates from CO₂ dissociation.

3. The reviewer wonders whether the stabilization process of CO₂^{•-} radicals is the RDS for the CO₂RR. Although the authors mentioned in the introduction that the existence of CO₂^{•-} radicals on metal electrodes, such as Au, Cu, and Ag, has been proposed based on the experimental Tafel slope analysis, experimental evidence and related discussions are missing in this work.

Response: The stabilization process of CO₂^{•-} radicals is one of the RDS for CO₂ reduction. As this step is the initial reaction, if it is kinetically non-favorable, the overall product yield of CO₂ transformation will be low. Therefore, this step is very important and decisive. In different catalytic systems, there are several RDS: initial electron transfer to CO₂ to form surface-bound CO₂^{•-} radicals, the first proton-coupled electron transfer, the following protonation of bound intermediates, or C-C coupling. These mechanisms have been reported mainly based on the experimental Tafel slope analysis (Rosen, J. *et al. ACS Catal.* **5**, 4293–4299 (2015).; Hatsukade, T. *et al. Phys. Chem. Chem. Phys.* **16**, 13814–13819 (2014).; Melchionna, M. *et al. Energy. Environ. Sci.* **14**, 5816–5833 (2021).; Chen, Y. *et al. J. Am. Chem. Soc.* **134**, 19969–19972 (2012).; Gao, S. *et al. Nature* **529**, 68–71 (2016).). Due to the large structural reorganization of the bent radical anion, the formation of the radical anion CO₂^{•-} by the first electron reduction occurs at very negative potentials, which is mainly recognized as the RDS for CO₂ reduction. We supplemented related discussions in the Introduction on P2 to P3 as the reviewer recommended.

4. Can the time-resolved measurements be performed at applied potentials related to CO₂RR. In this way, the time-resolved investigations is more meaningful for understanding the initial step for CO₂RR.

Response: We appreciate the suggestions. The use of working electrodes could be useful when the NPs are on the electrode and when the pulse radiolysis and electrochemistry are synchronized. Our laboratory previously combined pulse radiolysis with transient electrochemistry with a time resolution of 100

microseconds. We have published two papers on this subject, although they are in different fields (Zhou, X. *et al. Chem. Commun.* **52**, 251-263 (2016).; Latus, A. *et al. Chem. Commun.* **51**, 9089-9092 (2015).). So, we are well-versed in this area. However, as the time resolution is around 100 microseconds, the formation of bound $\text{CO}_2^{\bullet-}$ cannot be observed, and optical absorption cannot be used to follow the bound $\text{CO}_2^{\bullet-}$ on the electrode, which is not transparent. Instead of using the transient absorption spectrum as we did in the present work, the electrode could be used as a probe in some conditions to reoxidize the species formed by radiolysis or reduce it a second time. However, this is not the focus of the present fundamental study and does not contribute to the main findings on the binding reaction between $\text{CO}_2^{\bullet-}$ radicals and the NPs.

5. In Fig. 2, the authors discussed the effect of metallic sites on the stabilization process of $\text{CO}_2^{\bullet-}$ radicals. However, the surface chemical states are not considered since Cu and Ni are prone to be oxidized under current experimental conditions. Thus, the effect of surface oxidized species on surface-bound $\text{CO}_2^{\bullet-}$ radicals should be also included in the last section.

Response: We agree with the reviewer's comments. Indeed, the oxidized states of metals can influence the stabilization of $\text{CO}_2^{\bullet-}$ radicals. In the case of Ni and Cu NPs, we considered the oxidized sites. In the present study, all measurements are performed under reducing conditions (employing formate or tert-butanol as $\cdot\text{OH}$ scavenger), and the presence of oxidized sites is negligible. According to our results, the plasmon band of the NPs confirms the unoxidized states. In another study, we conducted experiments with oxidized sites and found oxidized copper forms different intermediates with $\text{CO}_2^{\bullet-}$ radicals exhibiting different kinetics compared to non-oxidized copper (unpublished results, please see Figure below). For these two reasons, we exclude the possibility that oxidized sites could play a major role in our conditions. Due to the length of this work, the oxidation state of copper deserves further in-depth investigation and discussion in future publications.

Figure R1. **a-b**, Transient kinetics at 350 nm within 850 μs (**a**) and transient absorption spectra (**b**) at 50 μs in 0.5 mM solution of Cu, Cu_2O , and CuO solutions.

6. For Fig. 3e, it is strongly suggested a direct comparison between the experimental data with the calculation results, as well as data analysis and discussion.

Response: We added the experimental results in Fig. 3e to see better the comparison. The experimental observations are limited to 300 nm (4.1 eV), and the simulations show the absorption spectra event at eV (250 nm).

Fig. 3 | The simulation of $\text{CO}_2^{\bullet-}$ radical stabilized on metal nanocatalysts. a-b, DFT optimized structures and DORI plots (repulsive forces in red and covalent interaction in blue) of $(\text{CO}_2^{\bullet-})_{\text{Cu}_m}^{\text{ad}}$ (a) and $(\text{CO}_2^{\bullet-})_{\text{Au}_m}^{\text{ad}}$ (b). **c-d,** electronic transition analysis (hole in blue and electron in green) of $(\text{CO}_2^{\bullet-})_{\text{Cu}_m}^{\text{ad}}$ (c) and $(\text{CO}_2^{\bullet-})_{\text{Au}_m}^{\text{ad}}$ (d). **e,** Transient absorption spectra (at 800 μs) and TD-DFT electronic spectra of $\text{CO}_2^{\bullet-}$ and $\text{CO}_2^{\bullet-}$ stabilized on nanocatalysts.

7. There are so many typos and grammar mistakes in the manuscript. For instance, line 206, page 11. The authors should carefully check them throughout the manuscript.

Response: We thank the reviewer. We did our best to correct the typos and grammar mistakes through the manuscript.

Reviewer #2 (Remarks to the Author):

I am not a specialist in radiolysis, but the experiment described in this paper is interesting and valuable. However, there are some major revisions needed.

The authors refer extensively to electrochemistry and electrocatalysis, but there is no electrochemistry or catalysis (or catalytic reaction) in their experiment, as far as I can see. The authors need to provide a more detailed description of how exactly their experiment relates to electrocatalysis (for instance, the issue of electrode potential). It is believed that under electrochemical conditions, the CO_2 radical cation is only formed at very negative potentials. How does that relate to the authors' experiments?

Response: We appreciate the comments. Radiolysis is often referred to as “an electrolysis process without electrodes”, which implies certain similarities. One may also view the high-energy radiation (X-rays/accelerated e^-) to represent one form of photons with sufficient energy to eject electrons directly from the water. Unlike electrochemical or photochemical reduction, where electrons are provided by the external potential or photo-induced carriers (e^- , h^+) separation in semiconductor materials, the radiation-driven approach produces electrons in bulk solutions via water ionization, well-known as the hydrated electron. The standard potential of the hydrated electron is -2.87 V vs SHE. Our previous reports have appreciated their contributions to water splitting and CO_2 reduction (Hu, C. *et al.* *J. Am. Chem. Soc.* **145**, 5578–5588 (2023)). In radiolysis, the first reduction step is thermodynamically favorable. The hydrated electron then reacts with CO_2 in solution and forms $\text{CO}_2^{\bullet-}$, followed by diffusion and reaction with NPs. ($e_{\text{aq}}^- + \text{CO}_2 \rightarrow \text{CO}_2^{\bullet-}$, $\text{CO}_2^{\bullet-} + \text{NP} \rightarrow (\text{CO}_2^{\bullet-})_{\text{NP}}^{\text{ad}}$). The final state, in which $\text{CO}_2^{\bullet-}$ is bound to the NPs, is equivalent to the product generated through either radiolysis or direct reduction of CO_2 on the electrode surface. In this study, we employ time-resolved spectroscopy measurements to investigate the ability of NPs to stabilize $\text{CO}_2^{\bullet-}$ on their surface. Notably, this is the first time that the optical

signature of $\text{CO}_2^{\cdot-}$ bound to NPs has been observed, enabling us to assess the reaction kinetics of these bound radicals.

As a result, the underlying process, though fundamentally different, would generate the same carbon-based intermediates as conventional chemical routines. At present, it is more important to clarify the reaction process of intermediates on metal surfaces at the beginning of the electrochemistry reduction process. This work focuses on the interaction of surface-bound $\text{CO}_2^{\cdot-}$ radicals decoupled the electric field. We emphasize the stabilization process and the subsequent decay process of $\text{CO}_2^{\cdot-}$ radicals on the metal surfaces. We added the discussion on P3.

Another important is the role of cations. The authors seem to have missed the recent discussions on the role of cations in CO_2 reduction, some literature claiming that cations are indispensable for CO_2 reduction. It is a pity that this paper does not relate to that discussion. In particular, an experiment with no cations (only protons) would be have been potentially extremely insightful. Entries into the literature: <https://www.nature.com/articles/s41929-021-00655-5>, <https://pubs.acs.org/doi/10.1021/jacs.2c11643>

Response: We thank the reviewer for the instructive comment to improve our understanding of the role of cations. We have added these references and discussions in the revised version. Additional experiments were performed by centrifugation twice to remove the sodium cations and by using formic acid to adjust the concentration of protons. Compared with the previous results containing 50 mM sodium ions, the kinetics of $\text{CO}_2^{\cdot-}$ radicals stabilizing on Au surfaces was slower, and the intensity of the absorption of surface-bound $\text{CO}_2^{\cdot-}$ radicals on Au NPs remarkably decreased (**Supplementary Fig. 14a-d**), which indicated that the adsorption process of $\text{CO}_2^{\cdot-}$ radicals on Au surfaces was undermined without metal cations. Moreover, transient absorption spectra revealed that the lifetime of surface-bound intermediates without metal cations was clearly curtailed regarding the sodium system (**Supplementary Fig. 14e-g**), suggesting that the primary role of cations is to stabilize the surface-bound $\text{CO}_2^{\cdot-}$ radicals. We have added them to the revised manuscript. Please find them on P17.

Supplementary Fig. 14 | No metal cation on $\text{CO}_2^{\cdot-}$ radical stabilization process in the presence of 0.5 mM Au under different pH conditions within 750 μs . a-b, Transient kinetics within 800 ns at 350 nm (a) and 420 nm (b). c-d, Transient kinetics within 800 μs at 350 nm (c) and 420 nm (d). e-g, Transient absorption spectra at different time under pH 2.8 (e), pH 3.4 (e), and pH 4.2 (g).

In the abstract, what is meant with the word “holistic”?

Response: We removed this word that means “complete”.

In the introduction, please avoid presenting CO₂ reduction as a way to mitigate CO₂ emission. The only way to mitigate CO₂ emissions is to not emit it.

Response: We thank the reviewer for the meticulous comment. The clarification of the concept is very necessary, and we corrected this sentence.

Reviewer #3 (Remarks to the Author):

This work contributes to the efforts to understand the CO₂ electroreduction reaction better. I find this work very interesting. Its strong part is the use of a time-resolved technique, an interesting objective (transient kinetics of the coverage of the CO₂•– radical adsorption, the coverage of the important effects of the catalyst metal, the nanoparticle size (NP), and the electrolyte cation, as well as the use of DFT to interpret their experimental data. However, I recommend the authors to address the following minor comments to make the manuscript publishable.

1. Line 40 The statement “The existence of CO₂•– on metal electrodes, such as Au, Cu, and Ag, has been proposed based on the experimental Tafel slope analysis.” Needs a reference

Response: Thanks for the suggestions, and we have added the references for the experimental Tafel slope analysis.

17. Hatsukade, T Rosen, J. *et al.* Mechanistic insights into the electrochemical reduction of CO₂ to CO on nanostructured Ag surfaces. *ACS Catal.* **5**, 4293–4299 (2015).
18. Kuhl, K. P., Cave, E. R., Abram, D. N. & Jaramillo, T. F. Insights into the electrocatalytic reduction of CO₂ on metallic silver surfaces. *Phys. Chem. Chem. Phys.* **16**, 13814–13819 (2014).
19. Melchionna, M., Fornasiero, P., Prato, M. & Bonchio, M. Electrocatalytic CO₂ reduction: role of the cross-talk at nano-carbon interfaces. *Energy. Environ. Sci.* **14**, 5816–5833 (2021).
20. Chen, Y., Li, C. W. & Kanan, M. W. Aqueous CO₂ reduction at very low overpotential on oxide-derived Au nanoparticles. *J. Am. Chem. Soc.* **134**, 19969–19972 (2012).
21. Gao, S. *et al.* Partially oxidized atomic cobalt layers for carbon dioxide electroreduction to liquid fuel. *Nature* **529**, 68–71 (2016).

2. Improve grammar/style: For example: change “free nanocatalyst solution” to “nanocatalyst-free solution”

Response: We thank the reviewer for the careful comment. We corrected several mistakes in the revised version.

3. Fig. 1 in the main text and all similar figures have unclear/incomplete captions: For example, the caption of Fig. 1 should mention that (a) is Transient kinetics at 350 nm within 700 ns "as a function of metal concentration", (b) and (d) should specify the metal concentration. The same is applicable to the analogous graphs

Response: We checked all the figures and supplemented the details of the captions.

4. Lines 131-133: “The slower kinetics observed in the Cu system than Au can be attributed to its lower affinity towards CO₂•– radicals or to the lower NP concentration resulting from the larger size of Cu (5.41 nm) than Au (1.71 nm).” This statement is questionable, as Cu is stronger interacts with CO₂ than Au doi:10.1021/jp074570y. The authors also have their own computational simulation results to verify whether this is true or not.

Response: Thanks for the reviewer's instructive comment. We have revised the descriptions. Our additional experiments at the longer time (ms to the second range) are obvious and in agreement with the comment. We added them in **Fig. 3** with the latest experiment to explain the difference in kinetics between Cu and Au NPs. Please find them on P10 to P12

5. Line 135 "...different kinetics of [CO₂•⁻] between Au and Cu suggests that the distinct stabilization reaction and structure may result in selective catalysis." This conclusion does not have any ground unless the authors normalize the reaction rate by the surface areas of NPs. It is unclear why the comparison in the main text is done for the NPs with different size, even though the authors have the same sizes (5.4 Cu and 5.3 Au nm). The same argument applies to the discussion of the NC size effect on the reaction kinetics on p.11

Response: Thanks to the reviewer for the constructive comment. We supplemented the comparison on the kinetics with the same size of Au and Cu (**Supplementary Fig. 6**). Within 80 μs, the kinetics and intensity of CO₂•⁻ radicals are almost comparable in the presence of Au and Cu NPs due to the nearly identical surface areas. However, after 80 μs, surface-bound CO₂•⁻ radicals exhibited accelerated decay on Cu NPs, yet still increased on Au NPs. These different kinetics between the same size of Au and Cu can demonstrate the distinct stabilization reaction, which may result in selective catalysis. With additional experiments that we performed at a longer time (until 900 ms), the difference in the behavior is observed (**Fig. 3**). Please find them on P8.

Supplementary Fig. 6 | **a-b.** Transient kinetics at 350 nm with 5.3 nm Au and 5.4 nm Cu within 80 μs (**a**) and 800 μs (**b**) in 0.5 mM Metal NP solution. **c-d.** Transient kinetics at 420 nm with 5.3 nm Au and 5.4 nm Cu within 80 μs (**c**) and 800 μs (**d**) in 0.5 mM metal NP solution.

6. Line 151, "...transient kinetics to the millisecond range" Should be changed to "submillisecond range"

Response: We have revised this sentence.

7. Line 161 "plasma band" should be changed to "electron plasma band"

Response: We revised this word and used the "surface plasmon absorption band".

8. On p. 10, the authors discuss the difference in the coordination of the $\text{CO}_2^{\bullet-}$ radical with Au and Cu clusters. The comparison with the lit data on the coordination of CO_2 on the bulk surfaces would be recommended;

Response: We have added the lit data on the coordination of CO_2 on the bulk metal surfaces to compare with our results.

45. Huber, H., Mcintosh, D. & Ozin, G. A. A metal atom model for the oxidation of carbon monoxide to carbon dioxide. The gold atom-carbon monoxide-dioxygen reaction and the gold atom-carbon dioxide reaction. *Inorg. Chem.* **16**, 975-979 (1977).
46. Ozin, G. A., Huber, H. & McINTOSH, D. Metal atom chemistry and surface chemistry: (carbon dioxide) silver, $\text{Ag}(\text{CO}_2)$. A localized bonding model for weakly chemisorbed carbon dioxide on bulk silver. *Inorg. Chem.* **17**, 1472-1476 (1978).
47. Mascetti, J. & Tranquille, M. Fourier transform infrared studies of atomic Ti, V, Cr, Fe, Co, Ni, and Cu reactions with carbon dioxide in low-temperature matrices. *J. Phys. Chem.* **92**, 2177-2184 (1988).

9. Lines 238-239: the recommended citations relevant to the effect of cation on the interfacial reactivity of CO_2 are doi:10.1039/C8CP07807F and doi:10.1038/s41929-021-00655-5

Response: We added these references in the revised version.

51. Monteiro, M. C. O. *et al.* Absence of CO_2 electroreduction on copper, gold and silver electrodes without metal cations in solution. *Nat. Catal.* **4**, 654–662 (2021).
52. Chernyshova, I. V. & Ponnurangam, S. Activation of CO_2 at the electrode-electrolyte interface by a co-adsorbed cation and an electric field. *Phys. Chem. Chem. Phys.* **21**, 8797–8807 (2019).

10. How the spin state of the radical changes upon adsorption? Is it still radical?

Response: We are very grateful for this question from the reviewer. Indeed, we have carried out additional simulations, which have been very conclusive and in line with our experimental observations.

The spin state of the system is fixed during the DFT calculation. We used the spin state that corresponds to a $\text{CO}_2^{\bullet-}$ radical. However, at the end of the energy calculation, the single electron could be either still localized on the $\text{CO}_2^{\bullet-}$ or delocalized on the nanoparticle. The results depend on the nature of the chemical interaction between the $\text{CO}_2^{\bullet-}$ and the nanoparticle.

We have calculated the total spin density of the systems of the ground state at the same level of theory, which is defined as the difference between the electron density of the alpha electrons minus the electron density of the beta electrons (**Supplementary Fig. 10**). For gold, the spin density remains on the $\text{CO}_2^{\bullet-}$ so that it remains a radical species because the interaction is weakly covalent. For copper, the result is different, showing a large delocalization of the spin density as a sign of the stronger covalent interaction between $\text{CO}_2^{\bullet-}$ and the Cu nanoparticle.

Please find the revised part on P13.

Supplementary Fig. 10 | Total spin density of the ground state for of $(\text{CO}_2^{\bullet-})_{\text{Au}_m}^{\text{ad}}$ (left) and $(\text{CO}_2^{\bullet-})_{\text{Cu}_m}^{\text{ad}}$ (right).

Reviewer #4 (Remarks to the Author):

In this work, the authors performed time-resolved visible spectroscopy of CO₂/metal nanoparticle systems, following a radiolysis pulse that induced CO₂^{•-}- radical formation. The authors claim that they observed the transient behavior of the surface-bound CO₂^{•-}- radical, which is thought to play a key role in electrocatalytic CO₂ reduction. They explored the kinetics as a function of nanoparticle size and electrolyte conditions. Overall, this work appears technically solid. The manuscript is mostly well-written. However, I do have significant concerns in regards to the interpretation of the data and the transferability of the insights to catalytic systems:

(1) It is unclear if the observed transients attributed to surface-bound CO₂^{•-} are simply due to electron transfer to the nanoparticles (NPs). Au and Cu NPs exhibit significant absorbances at wavelengths below 600 nm. Therefore, electron transfer to these NPs from a species other than CO₂^{•-}- could give rise to the observed transient absorbances and altered kinetics (e.g., Fig. 1b). The authors should conduct a control experiment with a system that contains all components but CO₂. This is a critically important control experiment; in the absence of such an experiment, their transients cannot be conclusively attributed to CO₂^{•-}-NP adducts.

Response: In our first version, we showed that if we remove the NPs from the solutions (by centrifugation), the new absorption band is absent. This control experiment allows us to assign to new transient formed via the interaction between CO₂^{•-} radicals and NPs. As requested by the reviewer, we performed additional experiments in the absence of CO₂ (replaced by Ar). In these conditions, the only reducing species is the hydrated electron (tert-butanol as •OH scavenger forms a non-reactive radical). The results clearly show no absorption band as before with CO₂. The hydrated electron decays rapidly by the well-known reaction ($e_{\text{aq}}^- + e_{\text{aq}}^- \rightarrow \text{H}_2 + \text{OH}^-$). The results of these supplementary control experiments with others (Observing the formation of the new absorption band by varying the concentration and size of NPs reported in the manuscript) show very clearly that the new absorption band observed in the visible range is due to the interaction between CO₂^{•-} and Au/Cu NPs. In the revised version, we added the results in **Supplementary Fig. 3 f-g**, showing our results in the absence of CO₂. Please find them on P6.

Supplementary Fig. 3 | Transient kinetics and absorption spectra of CO₂^{•-} radical Ar-saturated or CO₂-saturated solutions with 0.5 mM Au NPs. a-b, Transient kinetics at 350 nm within 80 μs (a) and transient absorption spectrum at 80 μs (b).

(2) Assuming that the authors can confirm that the transients arise from CO₂^{•-}-NP adducts through appropriate control experiments, it is unclear to me how the reported information can be transferred to catalytic systems: (a) In an electro- or photocatalytic system, CO₂ adsorbs on the surface. Electron transfer to CO₂ occurs either concurrent with the adsorption or follows initial physisorption. In the

system investigated by the authors, $\text{CO}_2^{\bullet-}$ is first prepared in solution, which then adsorbs on the NP. So, the pathway is fundamentally altered. This strongly limits the transferability of the results to actual catalytic systems. For example, the authors investigated cation effects on the $\text{CO}_2^{\bullet-}$ kinetics. However, they found that $\text{CO}_2^{\bullet-}$ forms adducts with cations in solution, before $\text{CO}_2^{\bullet-}$ adsorption. Obviously, this pathway is not operational in a catalytic system. For this reason, I question the broader impact of the study.

Response: We agree with the reviewer. The generation pathway of $\text{CO}_2^{\bullet-}$ is fundamentally different from electrochemical or photocatalytic systems. We viewed that the reduced state ($\text{CO}_2^{\bullet-}$) bound on the NPs remains the same, independent of the generation pathway. The spectral characterization of this initial transient is important because it provides electronic insights into how the charge is delocalized on the well-defined NPs and suggests the interacting force when $\text{CO}_2^{\bullet-}$ is formed on the surface. To address this issue, most studies have been conducted by theoretical methods. Until now, such experimental measurements have been lacking mainly due to technical challenges. Besides, catalytic CO_2 reduction by hydrated electrons was also reported and appreciated by photochemical or plasma-driven and radiolytic approaches. Moreover, pulse radiolysis avoids the perturbations of electrodes and light absorbers, which often represent the necessary influencing factors that increase the complexities. As a result, our spectral data provided benchmark observations that could be transferred to validate the theories developed in the electro- or photocatalytic system. The result will not block the transferability to actual catalytic systems.

The equilibrium between $\text{CO}_2^{\bullet-}$ and the cation also exists when $\text{CO}_2^{\bullet-}$ radicals are bound on the surface of NPs. It is clear that, for the first step, the binding is not the same for both methods; however, the effect of the presence of the cation exists for both cases. We also performed additional measurements requested by Reviewer 1. Without cations, compared with the previous results, the kinetics of $\text{CO}_2^{\bullet-}$ radicals stabilizing on Au surfaces was retarded, and the absorption of surface-bound $\text{CO}_2^{\bullet-}$ radicals on Au NPs remarkably decreased (**Supplementary Fig. 14**), which indicated that the stabilization process of $\text{CO}_2^{\bullet-}$ radicals on Au surfaces was undermined without cations. We have added them to the revised manuscript, and please find them on P16 to P17.

(b) Once $\text{CO}_2^{\bullet-}$ has formed on the surface, the technique can – in principle – characterize its decay kinetics. However, it is unclear to me what the extracted kinetics mean in a broader context. That a metallic surface stabilizes the $\text{CO}_2^{\bullet-}$ radical is not surprising. I assume that statistically, only one $e^-/\text{CO}_2^{\bullet-}$ reaches a given NP. So the reaction does not further proceed. A more impactful experiment would follow-up with a second radiolysis pulse to convert the surface-adsorbed $\text{CO}_2^{\bullet-}$ radical to CO . The conversion kinetics of this step would be important to characterize, but it is not done in this study.

Response: Thanks for the reviewer's instructive comments. Indeed, it is not surprising to stabilize $\text{CO}_2^{\bullet-}$ radicals on metallic surfaces. However, besides observing transient intermediates, we also found the diverse electron distribution and selective kinetics of $\text{CO}_2^{\bullet-}$ radicals on different metal catalysts during the process.

In our conditions, there is more than one $\text{CO}_2^{\bullet-}$ radical bounding on the surface of a given NP. The amount of $\text{CO}_2^{\bullet-}$ produced by hydrated electrons in solutions, in all cases, is much larger than the concentration of the NPs. The ratio of $\text{CO}_2^{\bullet-}/\text{NP}$ depends on the size and concentration of the NPs and the competitive reaction of $\text{CO}_2^{\bullet-}$. That means each NPs can have several $\text{CO}_2^{\bullet-}$ on the surface, and the reaction between two $\text{CO}_2^{\bullet-}$ can also occur on the surface of the NPs. However, by pulse radiolysis, it is impossible to reduce CO_2 two times because, in the solution, we produce a pulse of the hydrated electron, which forms $\text{CO}_2^{\bullet-}$ and then we observe the stabilization of $\text{CO}_2^{\bullet-}$ on the surface of NPs. Therefore, we adapted our setup to supplement the sub-second timescale pulse radiolysis as the reviewer instructed. This part was performed with the contribution of a new author (Dr Sergey Denisov). Due to the setup of the streak camera for measuring the transient absorption, we can only observe the absorption spectra at a specific time (such as 50 ms, 150 ms, 500 ms, or 900 ms) rather than the continuous kinetics, then

we succeed in observing the next step, the reaction of surface-bound $\text{CO}_2^{\cdot-}$. In the case of Cu NPs, we observe the formation of another intermediate following the decay of surface-bound $\text{CO}_2^{\cdot-}$, which has an extended absorption from 300 nm to 600 nm. This new intermediate is stabilized up to 900 ms (**Fig. 3a**). According to the principle of pulse radiolysis, we can assume that the new intermediate is formed either by a reaction between the surface-bound $\text{CO}_2^{\cdot-}$ and another radical (or CO_2) occurring on the surfaces of Cu NPs. We exclude the formation of the steady-state stable product, such as oxalate, formate, or CO, due to the observed broad absorption in the visible. The numerical simulations exclude the formation of $(\text{CO}_2)_2^{\cdot-}$. To know the exact nature of this new intermediate, it needs much more simulations and experiments. As for Au NPs with similar size (and also smaller sizes of NPs), only the decay of surface-bound $\text{CO}_2^{\cdot-}$ was observed with no further reaction occurring (**Fig. 3b**). The behavior of surface-bound $\text{CO}_2^{\cdot-}$ on AuNP is very different from that on Cu NPs.

Besides, Au and Cu NPs showcased essential distinctions in the kinetics of surface plasmon absorption bands, implying the different nature of their surface-bound interaction. Within 1 ms, both Au and Cu NPs present bleaching of their surface plasmon absorption band due to the charge of surface-bound $\text{CO}_2^{\cdot-}$ radicals, revealing the stabilization process as discussed above. At a longer time (a few hundred ms), once the next-step reaction happens, and in the case of Cu NPs, a second intermediate species is observed without noticeable bleaching of the absorption band. Nevertheless, in the case of Au NPs, the bleaching of the surface plasmon absorption band persisted up to 900 ms, accompanied by the total decay of $(\text{CO}_2^{\cdot-})_{\text{Au m}}^{\text{ad}}$ radicals, which demonstrated the primary occurrence of individual charge transfer from surface-bound $\text{CO}_2^{\cdot-}$ radicals to AuNP rather than covalent interaction between Au NPs and $\text{CO}_2^{\cdot-}$ radicals (**Fig. 3d-e**). The different behavior of surface-bound $\text{CO}_2^{\cdot-}$ demonstrated the nature of Cu and Au for $\text{CO}_2^{\cdot-}$ stabilization and conversion, which can explain the selective catalysis property of Cu for CO_2 reduction.

Please find the revision on P10 to P12.

Fig. 3 | Second timescale absorption profiles. a-c, Transient absorption spectra of $(\text{CO}_2^{\cdot-})_{\text{NP}}^{\text{ad}}$ radicals at 1 ms, 50 ms, 150 ms, 500 ms, and 900 ms in the presence of 0.5 mM 5.4 nm Cu (a), 5.3 nm Au (b) and 1.7 mM 1.7 nm Au (c) nanoparticles. d-e, The curves of $\Delta\text{O.D.}$ versus time delay at 350 nm (d) and 540 nm (e).

(3) The authors consistently refer to the intermediate at $\text{CO}_2^{\cdot-}$. They determined that it persists on timescales of 100s of microseconds. In an aqueous solution, it appears unlikely that the radical persists on such long timescales. Although it is generally accepted that e- transfer precedes H+ transfer for this first step of CO_2 reduction, protonation of the radical is expected to be very fast. The authors should

discuss this possibility in the manuscript. In this regard, the language of the stabilization process is diffuse. For example, on p. 7, the authors note that “the coordination state progressively orientates and sustains for more than 80 microseconds”. It is unclear what this stabilization process entails.

Response: We agree with the reviewer. In the revised version, we changed the wording. We observed that the shape of the absorption band is slightly changing when $\text{CO}_2^{\bullet-}$ is bound to the NPs. That reveals a reorganization of $\text{CO}_2^{\bullet-}$ on the surface during the first 100 microseconds. However, after a few 100 microseconds, the shape of the absorption band does not change with time, showing that $\text{CO}_2^{\bullet-}$ is well stabilized on the surface of the NPs. This stabilization can also be accompanied by protonation.

Please find them on P8.

Supplementary Fig. 4 | Transient absorption spectrum within 80 μs in 0.5 mM of Au (left) and Cu (right).

(4) On p. 7, the authors note that “the slower kinetics observed in the Cu system than Au can be attributed to its lower affinity towards $\text{CO}_2^{\bullet-}$ radicals”. Though this is suggested as one of two possibilities, it appears an unlikely one. I would expect Au to be generally less reactive than Cu (few things adsorb on Au strongly). The authors statement should be properly qualified. Further, the authors may want to acknowledge other possibilities. For example, is it possible that the $\text{CO}_2^{\bullet-}$ radical on Cu decays to another species faster than on Au?

Response: We agree with the reviewer. Here, the slower kinetics observed in the Cu system than Au should be attributed to the lower NP concentration due to the larger size of Cu (5.41 nm) than Au (1.71 nm). We supplemented the comparison on the kinetics with the same size of Au and Cu (**Supplementary Fig. 6**). Within 80 μs, the kinetics, and intensity of $\text{CO}_2^{\bullet-}$ radicals are almost comparable in the presence of Au and Cu NPs due to the almost identical surface areas. However, after 80 μs, surface-bound $\text{CO}_2^{\bullet-}$ radicals exhibited accelerated decay on Cu NPs, yet still increased on Au NPs.

Furthermore, according to the supplementary subsecond results, we proposed that the surface-bound $\text{CO}_2^{\bullet-}$ on Cu NPs decays to another species faster than on Au. We observe the formation of another intermediate on Cu following the decay of $\text{CO}_2^{\bullet-}$ radicals at 0.05 s, but no further reaction occurred on Au NPs (**Fig. 3**), indicating the discrepancy nature of Cu and Au for $\text{CO}_2^{\bullet-}$ stabilization and conversion.

Please find them on P8 and P10 to P12.

(5) The sentences on p. 9 are confusing. The authors first discuss the “plasma band”, then they refer to the same band as the “plasmon band”. This should be corrected. This is also the only time in the manuscript that they use this terminology. It would be better to introduce this term on p. 4 when discussing Figure S1 (or avoid it altogether). I also recommend that in the sentence “We suggest that the rise of the bleaching signal with Au content, ...”, Figure 2g should be referenced. Response: We have revised the sentences.

(6) The actually data in the figures are difficult to see. Typically, only the fit to the data can be clearly seen. The authors should show the experimental data points more clearly (use more intense colors). Also, it is unclear in Fig. 1b, f, and j if the data shown in these 3D figures represent a fit to the data or the actual data (same for Figure S1). This should be clarified. Judging from some of the SI figures, it

appears that the spectra are much noisier than shown in these graphs. I also note that while the 3D perspective view looks nice, it makes it impossible to compare the amplitudes across spectra. Response: We have revised all the Figures to show experimental data more clearly. Moreover, the data in 3D figures are the fitted ones. We will clarify it in the revised version.

(7) There are some minor language problems throughout the manuscript. I recommend that the authors thoroughly read it and correct any mistakes. Examples: The plural “s” is missing in “nanocatalysts” in the title; p. 5: “we performed the control experiments” should read “we performed control experiments”; p. 10: “simulations are performed” should read “simulations were” performed.

Response: We agree with the reviewer and did our best in the revised version.

REVIEWER COMMENTS

Reviewer #1 (Remarks to the Author):

Although the authors have carried out extensive work to improve the quality of their study, some issues are still not well addressed.

1. In the introduction, the authors claimed that "Due to the large structural reorganization of the bound radical anion, the radical anion $\text{CO}_2^{\bullet-}$ formed by the first electron reduction occurs at very negative potentials, recognized as the critical RDS for CO_2 reduction". This description is arbitrary without any experimental evidence, and it is not appropriate to elucidate catalytic origins of different kinds of catalysts, such as Cu, Au and Ni in this work.
2. The authors are strongly suggested to provide Tafel analysis for the model catalysts (Cu, Au and Ni) rather than cite previous results in literatures in order to validate that the stabilization process of $\text{CO}_2^{\bullet-}$ radicals is the RDS for the CO_2RR .
3. In addition to Cu, transient kinetics and transient absorption spectra of Au and Ni should be provided in the SI (Fig. R1), and moreover, a concise discussion about the roles of oxidized species should be also included.
4. The authors highlights the significant roles of stabilization of $\text{CO}_2^{\bullet-}$ radicals for the CO_2RR , it is surprising that there is no related electrochemical CO_2RR testing on the Au, Cu and Ni and its correlation with the stabilization of $\text{CO}_2^{\bullet-}$ radicals. Unfortunately, following the authors' answers, I can only reconfirm the weakness of this work, but this is a key point for accepting a paper in a journal of this calibre.

Reviewer #2 (Remarks to the Author):

I am satisfied with the changes

Reviewer #3 (Remarks to the Author):

I am satisfied with the response to my comments

Reviewer #4 (Remarks to the Author):

The authors have thoroughly addressed my questions. They have conducted additional experiments that strengthen the work and have appropriately edited the manuscript. I estimate that the work will be of broad interest to the electrocatalysis community. I recommend acceptance of the manuscript in its present form.

Point-by-point responses to the reviewers' comments

(Manuscript ID: NCOMMS-23-18719-A)

Reviewer #1:

Although the authors have carried out extensive work to improve the quality of their study, some issues are still not well addressed.

Response: We thank the reviewer for the careful review and insightful comments. The issues have been addressed by supplementary electrochemical measurements. The results nicely matched our time-resolved findings and supported the conclusion.

1. In the introduction, the authors claimed that “Due to the large structural reorganization of the bound radical anion, the radical anion $\text{CO}_2^{\cdot-}$ formed by the first electron reduction occurs at very negative potentials, recognized as the critical RDS for CO_2 reduction”. This description is arbitrary without any experimental evidence, and it is not appropriate to elucidate catalytic origins of different kinds of catalysts, such as Cu, Au and Ni in this work.

Response: We appreciate the comment. The statement has been based on theoretical calculations and suggested by electrochemical studies. The radical anion $\text{CO}_2^{\cdot-}$ has been thought to be a key intermediate during catalytic CO_2 reduction. However, as the reviewer pointed out, direct experimental evidence has been indeed lacking. Recently, we showed that CO_2 -to-format and CO_2 -to- CH_3OH conversion under irradiation conditions starts with the formation of short-lived $\text{CO}_2^{\cdot-}$ in bulk solutions. We have revised this description and conducted additional experiments as below to elucidate the RDS of different kinds of catalysts in the work.

2. The authors are strongly suggested to provide Tafel analysis for the model catalysts (Cu, Au and Ni) rather than cite previous results in literatures in order to validate that the stabilization process of $\text{CO}_2^{\cdot-}$ radicals is the RDS for the CO_2RR .

Response: We thank the reviewer for the comment. We supplemented the Tafel analysis on the model catalysts prepared in our work. As shown in Figure below, the obtained Tafel slopes for CO production catalyzed by Cu, Ni, and Au nanoparticles are $179.4 \text{ mV dec}^{-1}$, $343.1 \text{ mV dec}^{-1}$, and $146.2 \text{ mV dec}^{-1}$, respectively. These results with our NPs agree with those already published in the literature (*J. Am. Chem. Soc.* 2012, 134, 4, 1986–1989; *ACS Energy Lett.* 2023, 8, 2185–2192; *Energy Environ. Sci.*, 2021, 14, 5816). In addition, the results indicated a rate-determining initial e^- transfer of adsorbed CO_2 to surface-bound $\text{CO}_2^{\cdot-}$ intermediates for the model catalysts in our work, excluding other possible RDS like protonation and coupling reactions of $\text{CO}_2^{\cdot-}$ intermediates.

Supplementary Fig. 3 | Tafel plots for CO production catalyzed by Cu, Ni, and Au.

3. In addition to Cu, transient kinetics and transient absorption spectra of Au and Ni should be provided in the SI (Fig. R1), and moreover, a concise discussion about the roles of oxidized species should be also included.

Response: We thank the reviewer for the comments. We have supplemented the transient kinetics and transient absorption spectra of Au and Ni in **Supplementary Fig. 5**.

The roles of oxidized species have been an important issue in revealing the design principle of catalysts and mechanistic understanding. We performed pulse radiolysis experiments on precisely controlled oxides-derived NPs. As previously mentioned in our response, we have paid close attention to the oxidation states of the particles and carried out measurements with copper particles containing Cu(I) and Cu(II) species.

The preliminary results indicated a significant effect of the oxidation state on the transient kinetics and absorption spectra. Therefore, the time-resolved spectral changes could correlate with the elementary process occurring during CO₂ reduction. For example, the coverage dynamic of CO₂⁻ radicals and the subsequent reactions on oxides-derived Cu NPs are very different from those with Cu(0) NPs. This forthcoming study is now in progress and will be the subject of future publication. For the information of the reviewer, we have included unpublished Figure R2 below, presenting the absorption spectra and transient kinetics of CO₂⁻ radicals on the surface of Cu(0), Cu(I), and Cu(II) NPs. We have also included a concise discussion about roles.

Fig. R1 (Supplementary Fig. 5 in the manuscript) | a-c, Transient absorption spectrum within 80 μs in 0.5 mM of Au (a), Cu (b) and Ni (c). d-f, Transient kinetics at 350 nm within 80 μs in the presence of different concentrations of Au (d), Cu (e), and Ni (f).

Fig. R2 Reactivity and electron structure of surface-bound $\text{CO}_2^{\bullet-}$ radicals on Cu(0), Cu(I), and Cu(II). (A to C) Transient absorption matrix within 820 μs in CO_2 -saturated 0.1 M formate solution in the presence of 0.5 mM Cu(0) (A), Cu(I) (B), Cu(II) (C) NPs. (D to F) Transient absorption spectra at 20 μs for Cu(0) (D), Cu(I) (E), Cu(II) (F) NPs. (G to I) Transient kinetics within 800 μs for Cu(0) NPs at 420 nm (G), Cu(I) NPs at 370 nm (H), Cu(II) NPs at 380 nm (I) Inset: The fitting kinetics curves of second-order reaction.

4. The authors highlights the significant roles of stabilization of $\text{CO}_2^{\bullet-}$ radicals for the CO_2RR , it is surprising that there is no related electrochemical CO_2RR testing on the Au, Cu and Ni and its correlation with the stabilization of $\text{CO}_2^{\bullet-}$ radicals. Unfortunately, following the authors' answers, I can only reconfirm the weakness of this work, but this is a key point for accepting a paper in a journal of this calibre.

Response: We have performed the electrochemical CO_2RR testing on the Au, Cu, and Ni. The data demonstrate the correlation of different catalysis performances with the stabilization kinetics of $\text{CO}_2^{\bullet-}$ radicals. According to our electrochemical results, Cu induced the formation of CO, CH_4 , and C_2H_4 . In contrast, Ni predominately produced hydrogen with much lower activity to reduce CO_2 , and Au mainly produced CO from the conversion of CO_2 (Supplementary Fig. 11).

The tendency of electrochemical CO_2RR testing agrees with our pulse radiolysis observation. We found that Ni NPs cannot stabilize $\text{CO}_2^{\bullet-}$ radicals, which corroborated its slight activity of CO_2 reduction; however, Cu and Au NPs can stabilize $\text{CO}_2^{\bullet-}$ radicals. The long-lived characteristic spectra disclosed that the stabilization process of $\text{CO}_2^{\bullet-}$ radicals on the surface of Au and Cu has substantially extended the lifetime of $[\text{CO}_2^{\bullet-}]_{\text{ad}}$ radicals by at least 100 times compared to $\text{CO}_2^{\bullet-}$ radicals in solutions or in the presence of Ni. The spectral observations lay the groundwork for the subsequent multi-electron transfer reaction.

Moreover, the differentiation in the respective $[\text{CO}_2^{\cdot-}]_{\text{ad}}$ radical transient kinetics and characteristic spectra across Au, Cu, and Ni systems suggest diverse stabilization behavior and adsorbed structures of $[\text{CO}_2^{\cdot-}]_{\text{ad}}$ radicals on various metal surfaces. This selective stabilization process determines the subsequent selective reduction pathway of $[\text{CO}_2^{\cdot-}]_{\text{ad}}$ radicals. Notably, we also observed the dimerization pathway at a subsecond range of surface-bound $\text{CO}_2^{\cdot-}$ radicals on Cu rather than Au, which is in accordance with the unique property of Cu to produce C_2H_4 .

Supplementary Fig. 11 | Partial current density for different products by Cu(a) and Ni (b)

Reviewer #2:

I am satisfied with the changes

Response: We thank the reviewer for the positive comment and careful review of our manuscript.

Reviewer #3:

I am satisfied with the response to my comments

Response: We thank the reviewer for the careful review of our manuscript and the recommendation to publish our work.

Reviewer #4:

The authors have thoroughly addressed my questions. They have conducted additional experiments that strengthen the work and have appropriately edited the manuscript. I estimate that the work will be of broad interest to the electrocatalysis community. I recommend acceptance of the manuscript in its present form.

Response: We appreciate the reviewer for the recommendation and previous instructive comments. We look forward to communicating with the electrocatalysis community to further reveal the unsolved mechanism of electrochemical CO₂ reduction.

REVIEWERS' COMMENTS

Reviewer #1 (Remarks to the Author):

The authors have supplemented experiments for supporting their conclusions and addressed my questions. I recommend acceptance of the manuscript in its present form.

Point-by-point responses to the reviewers' comments

(Manuscript ID: NCOMMS-23-18719-B)

Reviewer #1:

The authors have supplemented experiments for supporting their conclusions and addressed my questions. I recommend acceptance of the manuscript in its present form.

Response: We thank the reviewer for the careful review of our manuscript and the recommendation to publish our work.